# Rethinking Visual Intelligence: Insights from Video Pretraining

## Abstract

Large language models (LLMs) have demonstrated that large-scale pretraining enables systems to adapt rapidly to new problems with little supervision in the language domain. This success, however, has not translated as effectively to the visual domain, where models, including LLMs, continue to struggle with compositional understanding, sample efficiency, and general-purpose problem-solving. We investigate Video Diffusion Models (VDMs) as a promising direction for bridging this gap. Pretraining on spatiotemporal data endows these models with strong inductive biases for structure and dynamics, which we hypothesize can support broad task adaptability. To test this, we design a controlled evaluation in which both a pretrained LLM and a pretrained VDM are equipped with lightweight adapters and presented with tasks in their natural modalities. Across benchmarks including ARC-AGI, ConceptARC, visual games, route planning, and cellular automata, VDMs demonstrate higher data efficiency than their language counterparts. Taken together, our results indicate that video pretraining offers inductive biases that support progress toward visual foundation models.

## 1 Introduction

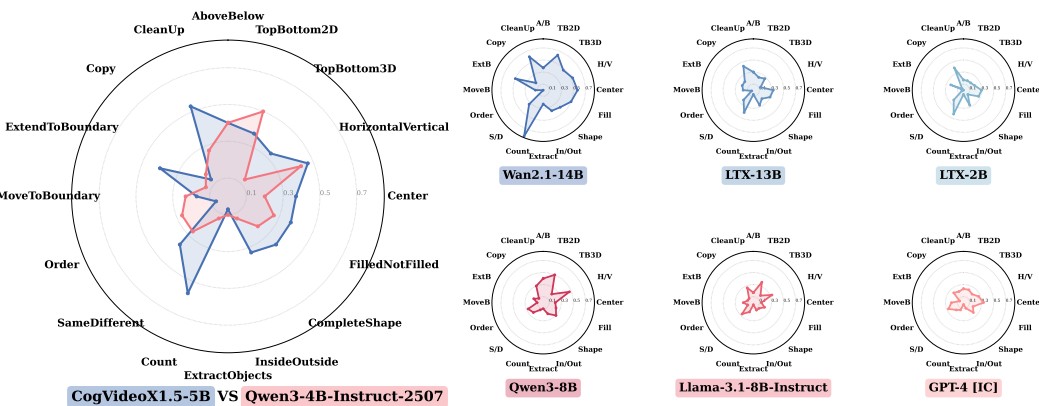

Figure 1: Radar plot showing ConceptARC competencies between **VDMs** and **LLMs**, GPT-4 [IC] is added for additional reference.

Foundation models have reshaped natural language processing by showing that large-scale pretraining can equip models with broad knowledge and strong inductive priors. This foundation allows models to adapt quickly and effectively to new tasks through techniques like in-context learning (Brown et al., 2020) and parameter-efficient fine-tuning (Liu et al., 2022), achieving strong performance with minimal supervision. The success of Large Language Models (LLMs) illustrates how scale and pretraining can create systems that generalize across diverse problems. Achieving a similar level of versatility in vision, however, remains largely unexplored and a major challenge. Despite recent breakthroughs in image and video generation (Labs, 2025; Polyak et al., 2024; Qin et al., 2024), vision models are not yet on par with LLMs when it comes to compositional skills, sample efficiency, and versatility in problem solving.

Video Diffusion Models (VDMs) represent an exciting direction for narrowing this gap. Pretraining on rich spatiotemporal data endows them with strong inductive biases for spatial structure and temporal dynamics (Blattmann et al., 2023; Google DeepMind, 2025; Wu et al., 2025), which we hypothesize can be harnessed for structured visual understanding. We move beyond treating videos as mere generative artifacts and instead regard them as a natural representation for problem solving, where tasks are expressed as transformations unfolding over time. Building on this perspective, we introduce a simple and general framework for adapting VDMs to a broad class of visual tasks and evaluate them head-to-head with equally adapted LLMs (see Figure 1). This setup allows us to test whether large-scale video pretraining offers a complementary foundation for structured visual problem-solving, contrasting the strengths of visually grounded models with those of symbolically trained language models.

Each task is represented consistently but adapted to each model family's modality: LLMs operate in a text-to-text setting, where inputs and outputs are serialized into structured text, while VDMs receive an image-to-image formulation, where input–output pairs are rendered as short videos to model the task as a temporal transformation. Both model families use identical LoRA-based Hu et al. (2022) adaptation: adapters are inserted at corresponding layers, pretrained backbones remain frozen, and only lightweight parameters are updated. This symmetry provides a controlled basis for comparison and isolates the impact of video pretraining on structured visual understanding.

Our contributions are as follows:

1. A unified framework for adapting VDMs to image-to-image visual tasks by reframing examples as temporal sequences.

2. A controlled evaluation setting where both VDMs and LLMs are fine-tuned with LoRA-based adaptation, enabling direct comparison.

3. Empirical evidence that VDMs benefit from video pretraining for visual intelligence, hinting at a path toward flexible visual foundation models with both generative and problem-solving strengths.

## 2 RELATED WORK

**Language Foundation Models.** LLMs have demonstrated remarkable generalization and adaptability to new tasks with minimal supervision, mainly due to their large-scale pretraining on diverse text corpora Brown et al. (2020); Chowdhery et al. (2023). Their extensive pretraining equips LLMs with rich knowledge and strong inductive biases, enabling them to perform few-shot learning Brown et al. (2020) and in-context learning Coda-Forno et al. (2023), where models learn new tasks only by observing a handful of examples. Parameter-efficient finetuning methods like LoRA Hu et al. (2022) extend this adaptability allowing LLMs to specialize to new domains while the backbone is completely frozen Liao et al. (2025). Together, these capabilities make LLMs highly flexible and scalable problem solvers. In this paper, we leverage this adaptability to compare the data efficiency of LLMs and VDMs across diverse visual tasks.

**Video Diffusion Models.** Diffusion-based generative models have recently achieved remarkable progress in video synthesis. Pioneering approaches such as CogVideo Hong et al. (2022) and Villegas et al. (2022) introduced scalable architectures for text-to-video generation. More recent models like Sora Qin et al. (2024), MovieGen Polyak et al. (2024), Veo 3 Google DeepMind (2025), and CogVideoX Yang et al. (2024) set new standards for quality and realism. Recent work has investigated controllable video generation NVIDIA et al. (2025); Hassan et al. (2025); Kanervisto et al. (2025), with the goal of producing realistic, high-quality videos while allowing precise control over motion and dynamics. These methods emphasize modeling dynamic environments and predicting plausible future states conditioned on past observations and control inputs.

**Visual Foundation Models** Recent work has investigated the use of generative models as generalist vision models. Methods such as image inpainting for visual prompting Bar et al. (2022) and image-based in-context learning Wang et al. (2023a) demonstrate that structured inputs can enable these models to solve diverse tasks. Diffusion models have further been extended to in-context learning Wang et al. (2023b), instruction following across heterogeneous tasks Geng et al. (2024),

and broader computer vision problem solving Zhao et al. (2025). Sequential modeling has been proposed as a unified interface for scaling vision models Bai et al. (2024). Building on this line of work, Lin et al. (2025) train CogVideoX1.5 with temporal in-context prompts for multi-task learning, but their focus remains on broad computer vision benchmarks rather than visual intelligence, and their method requires extensive training[1].

Our approach does not attempt to build a foundation model from scratch. Instead, we investigate whether a pretrained VDM, pretrained extensively on next-frame prediction, can begin to exhibit the properties expected of visual foundation models by leveraging inductive biases gained through spatiotemporal pretraining.

## 3 METHODOLOGY

### 3.1 SETUP AND COMPARISON PROTOCOL

We adopt the definition of intelligence proposed by Chollet (2019):

> *The intelligence of a system is a measure of its skill acquisition efficiency over a scope of tasks with respect to priors, experience, and generalization difficulty.*

This perspective motivates our evaluation design. We focus not only on absolute accuracy but also on how quickly models acquire new capabilities when exposed to limited supervision.

To evaluate our hypothesis we curate a diverse benchmark of visually grounded tasks that can be specified textually as grid-based problems, including ARC-AGI, Sudoku solving, and route planning. We now describe the evaluation setup in detail.

Let $\mathcal{T}$ denote a task with dataset $\mathcal{D}_{\mathcal{T}} = \{(x_i, y_i)\}_{i=1}^{n}$, where each $x_i$ and $y_i$ is an input-output pair. Each sample is expressed in two complementary modalities:

**Image**  An **image pair** $(I(x_i), I(y_i))$, where $I(\cdot)$ deterministically renders RGB images of size $(3 \times H \times W)$.

**Text**  A **JSON pair** $(J(x_i), J(y_i))$, where $J(\cdot)$ maps a grid to a compact JSON string.

We serialize samples in a neutral format that avoids domain-specific priors, requiring both models to infer task rules directly from raw representations. Training and evaluation splits are identical across all models to ensure a fair and controlled comparison. VDMs are trained directly on the image modality using our approach, which we detail in the next section, while LLMs are trained on the text modality.

We define accuracy as the proportion of test instances where the predicted output *exactly matches* the ground truth grid. For tasks where multiple valid solutions may exist (e.g., *Sudoku*, *Sudoku Mini*, *Hitori*), we filter datasets to ensure each instance has an unique solution. When unique solutions cannot easily be guaranteed, as in *Shortest Path*, we introduce complementary metrics to better capture solution quality (see Section 4.2.2).

To evaluate efficiency of skill acquisition, we consider two complementary settings.

**ARC Family.** We evaluate models on ARC-AGI and ConceptARC, where the challenge is to solve diverse tasks from only 2–5 demonstrations. Following prior work Moskvichev et al. (2023); Chollet (2019); Li et al. (2025), we measure how many tasks each model can solve under this minimal supervision regime.

**Structured Visual Tasks.** We then turn to structured benchmarks. Here we systematically vary $n$, the number of training examples per task, to trace curves and quantify the rate of skill acquisition rather than focusing solely on endpoint accuracy.

---

[1]We add qualitative results on standard computer vision tasks in the Appendix to show that our framework can also be extended to this setting.

## 3.2 ADAPTING VIDEO DIFFUSION MODELS FOR IMAGE-TO -IMAGE

We adapt pretrained VDMs to image-to-image (I2I) prediction tasks by re-framing each input–output pair $(I_{x_i}, I_{y_i})$ as a short *transition video*. This leverages the generative prior of VDMs, while requiring minimal supervision.

**Transition video construction** Each pair $(x_i, y_i)$ is converted into a temporal sequence $v_i = [v_{i,1}, \dots, v_{i,F}]$, where

$$v_{i,1} = I(x_i), \quad v_{i,F} = I(y_i).$$

Intermediate frames are generated with an interpolation function $\phi$. For example, a *convex interpolation* produces a smooth transition

$$v_{i,f} = (1 - \alpha) I(x_i) + \alpha I(y_i), \text{ where } \alpha = \tfrac{f-1}{F-1}, \text{ and } f = 1, \dots, F,$$

while a *discrete interpolation* simply holds the input frame for the first half of the sequence and afterwards switches to the output frame:

$$v_{i,f} = \begin{cases} I(x_i), & f \le F/2, \\ I(y_i), & f > F/2. \end{cases}$$

This yields a dataset $\mathcal{V}_{\mathcal{T}}$ of input-conditioned video trajectories. For our comparisons, we adopt the *discrete interpolation* to avoid introducing any biases.

**Fine-tuning** We adapt a pretrained VDM by conditioning on the first frame $v_1^0$ and a neutral fixed text embedding $e_{\text{text}}$. Given a noisy video $v^t$ at step $t$, the model predicts noise $\epsilon_\theta$ via

$$\mathcal{L}_{\text{VDM}} = \mathbb{E}_{v^0 \sim \mathcal{V}_{\mathcal{T}}, \epsilon \sim \mathcal{N}(0,\mathbf{I}),t} \left[ \|\epsilon - \epsilon_\theta(v^t, t, c)\|_2^2 \right], \quad c = \{v_1^0, e_{\text{text}}\}.$$

We use LoRA modules for fine-tuning, updating only these additional weights while keeping the pretrained model frozen.

**Inference** At test time, the model generates predictions through reverse diffusion. The procedure is detailed in Algorithm 1.

This procedure reframes image-to-image prediction as a conditional video generation problem, enabling efficient adaptation of pretrained VDMs to new tasks.

## 3.3 ADAPTING LARGE LANGUAGE MODELS

We adapt pretrained LLMs to structured prediction tasks by framing each example as a JSON-to-JSON translation problem.

**Fine-tuning** We adapt pretrained LLMs using a standard sequence-to-sequence objective. Given tokenized input–output pairs, the model is trained to maximize the likelihood of the target sequence under teacher forcing:

$$\mathcal{L}_{\text{LLM}} = \frac{1}{n} \sum_{i=1}^{n} \sum_{t=1}^{|\mathbf{v}_i|} - \log p_\theta(v_{i,t} \mid \mathbf{u}_i, \mathbf{v}_i^{<t}).$$

We insert LoRA modules into the pretrained backbone, fine-tuning only these lightweight adapters while keeping the majority of parameters frozen.

**Inference** At test time, predictions are generated autoregressively. The procedure is summarized in Algorithm 2.

**Algorithm 1** Inference for VDM

1: Encode input: $c_{\text{test}} \leftarrow \{I(x_{\text{test}}), e_{\text{text}}\}$
2: Initialize noise: sample $v^T \sim \mathcal{N}(0, \mathbf{I})$
3: Reverse diffusion: recover $v^0$ conditioned on $c_{\text{test}}$
4: Output prediction: $\hat{y} \leftarrow v_F^0$ (final frame)

**Algorithm 2** Inference for LLM

1: Encode input: $J(x_{\text{test}})$ as JSON string
2: Tokenize and feed sequence into model
3: Autoregressively decode output until termination
4: Return prediction: $\hat{y}$ as JSON string

## 4 EXPERIMENTS

### 4.1 ARC FAMILY

The ARC-AGI benchmark Chollet (2019) evaluates an agent's ability to infer and apply abstract patterns through compositional understanding, few-shot learning, and inductive generalization. Each ARC task provides only a handful of input–output examples (typically 2–5), requiring the model to discover the underlying transformation rule and apply it to novel test inputs. This benchmark is widely regarded as a challenging measure of progress in abstraction and generalization.

We follow the evaluation protocol of Chollet et al. (2024), which allows up to two attempts per test input and counts a question as solved only if all predictions match the ground truth. Quantitative results appear in Table 1, with qualitative examples in Figure 3. For comparison, we also report single-attempt results of commercial LLMs from Chollet et al. (2024). Figure 2 illustrates the overlap between tasks solved by the VDM and the LLM, underscoring their complementary strengths.

Table 1: ARC-AGI test performance. Following the official evaluation protocol Chollet et al. (2024), models are evaluated with two attempts per test input. We also report single-attempt results for comparability with commercial LLMs, which are only available under this setting.

| Model | Accuracy (%) |
|---|---|
| **Two-attempt setting** | |
| CogVideoX1.5-5B | 16.75 |
| Qwen3-4B-Instruct-2507 | 8.00 |
| **Single-attempt setting** | |
| CogVideoX1.5-5B | 12.50 |
| Qwen3-4B-Instruct-2507 | 6.75 |
| OpenAI o1-preview | 21.00 |
| Anthropic Claude 3.5 Sonnet | 21.00 |
| OpenAI GPT-4o | 9.00 |
| Google Gemini 1.5 | 8.00 |

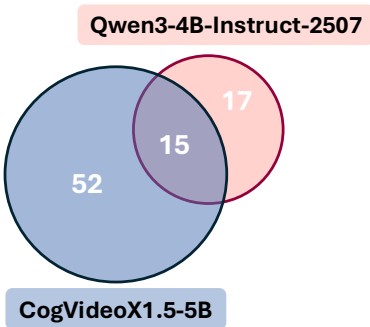

Figure 2: Venn diagram of ARC-AGI tasks showing those solved exclusively by each model and those solved by both.

We evaluate models on ConceptARC Moskvichev et al. (2023), a curated variant of ARC designed to systematically measure visual concept understanding and generalization. ConceptARC groups tasks into 16 concept categories (for example, Above and Below, Center, Count), with each category containing 10 tasks. Each task includes 3 distinct test inputs, creating controlled variation in visual patterns and object relationships while maintaining internal consistency within each concept group. Following the protocol of Moskvichev et al. (2023), we allow three attempts per test input and mark an input as solved if any attempt is correct. Performance is reported in Figure 1, where we further include as VDMs: Wan2.1-14B Wang et al. (2025), LTX-13B, LTX-2B HaCohen et al. (2025), CogVideoX1.5-5B Yang et al. (2024) and as LLMs: Qwen3-4B-Instruct-2057, Qwen3-8B Qwen3-4B-Instruct-2507 Team (2025), Llama3.1-8B Meta-AI (2024), and GPT-4 in an IC setting Moskvichev et al. (2023). Full table with results is included in the Appendix.

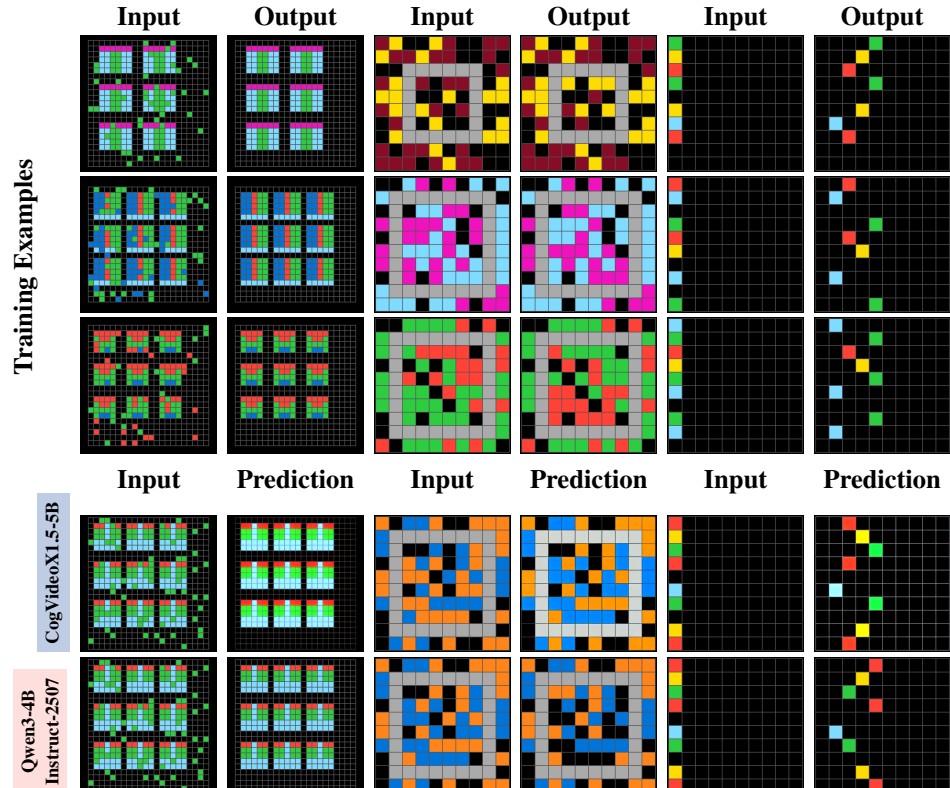

Figure 3: Qualitative results on ARC-AGI for problems *0607ce86*, *7ee1c6ea*, and *f45f5ca7*.

These results highlight the importance of strong visual priors: by leveraging representations that capture spatial structure, compositionality, and low-level visual cues, the VDM is able to approach these abstract tasks in a way that improves upon traditional text-centric approaches.

## 4.2 STRUCTURED VISUAL TASKS

From this point onward, we focus on one representative model from each family: CogVideoX1.5-5B Yang et al. (2024) for video diffusion models and Qwen3-4B-Instruct-2507 Qwen3-4B-Instruct-2507 Team (2025) for language models. This pairing aligns model scale while contrasting pretraining modalities, allowing us to examine how different priors influence adaptability to visually grounded tasks.

### 4.2.1 VISUAL GAMES

As part of our broader evaluation, we examine performance on a diverse set of five visual games that span both puzzle-solving and board play. These tasks provide an additional perspective on how the models handle structured visual inputs and varying interaction styles. The puzzle-based tasks, *Hitori 5x5* and two versions of *Sudoku* (standard one and *Mini*), focus on solving constraint-based problems in structured grids, where success depends on extracting spatial patterns and enforcing global consistency from local information. The board games, *Connect 4* and *Chess Mate-in-1*, shift attention to game scenarios where the goal is to identify the winning move in a given configuration. Together, these games cover a range of visual layouts and structured objectives, complementing the other tasks explored in this study.

Figure 4 presents model performance as a function of the number of training samples. CogVideoX1.5-5B demonstrates strong scaling behavior across most tasks, surpassing Qwen3-4B-Instruct-2507 in four of the five games. Its advantage is particularly clear in *Sudoku* and *Hitori*, which rely on interpreting complex grid layouts and visual compositions. This supports the view

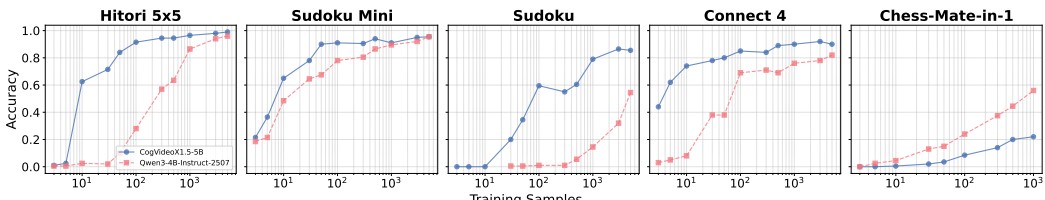

Figure 4: Accuracy as a function of training set size for **CogVideoX1.5-5B** and **Qwen3-4B-Instruct-2507** on five visual games.

that VDMs capture compositional features in visual data more effectively than LLMs, which are primarily optimized for language. The only exception is chess, where Qwen3-4B-Instruct-2507 performs better, likely reflecting the abundance of chess material in textual corpora that LLMs can partially internalize during pretraining Kuo et al. (2023).

### 4.2.2 ROUTE PLANNING

We evaluate route planning in 2D grid environments through two tasks: *Maze* and *Shortest Path*. In *Maze*, the model must navigate from the top-left to the bottom-right corner of a grid. In *Shortest Path*, the objective is to connect two arbitrary points with the shortest possible route. For *Shortest Path*, we report two complementary metrics to assess model performance:

**Path Success Rate (PSR)**   The percentage of evaluation examples where the predicted path forms a continuous connection between the source and target locations.

**Relative Path Length (RPL)**   For cases **where a valid path is produced**, we compute

$$\text{RPL} = \frac{\text{Predicted Path Length}}{\text{Ground-Truth Shortest Path Length}}.$$

This value may increase even as overall performance improves, since better models tend to predict good paths for more challenging cases, potentially constructing longer yet valid paths.

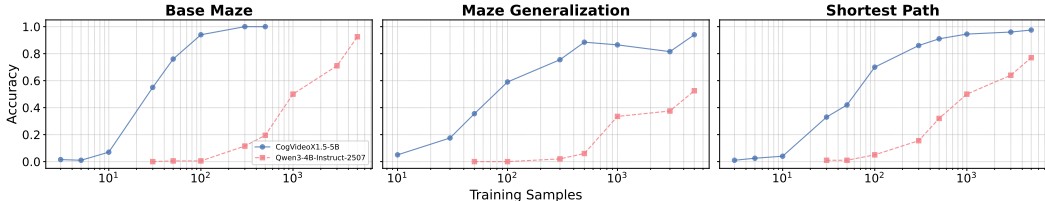

Figure 5: Accuracy as a function of training set size for **CogVideoX1.5-5B** and **Qwen3-4B-Instruct-2507** on *Base Maze*, *Maze Generalization*, and *Shortest Path*.

For *Maze*, we evaluate in two settings: a **matched-scale** (*Base Maze*) scenario, where both training and evaluation are conducted on $21 \times 21$ mazes to study performance as a function of training sample size; and a **generalization** scenario, where models are trained on smaller $13 \times 13$ grids and tested on larger $21 \times 21$ grids to assess cross-scale generalization (*Maze Generalization*).

Accuracy results are shown in Figure 5. For *Shortest Path*, additional metrics are reported in Table 2. The VDM consistently constructs valid paths with far fewer supervised examples, achieving up to a tenfold reduction in data requirements in low-sample regimes, which underscores its stronger inductive biases relative to the LLM. Moreover, it demonstrates the ability to generalize much quicker from limited training on smaller mazes to larger, more complex ones.

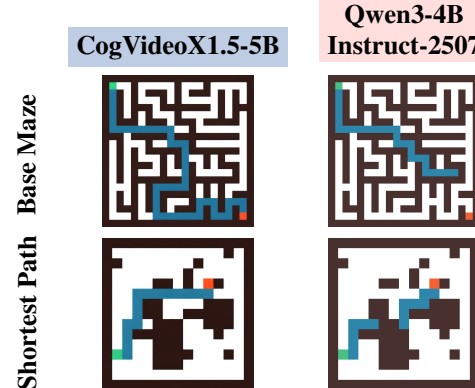

Figure 6: Qualitative examples for *Base Maze* and *Shortest Path* tasks, after fine-tuning with $n = 300$ samples.

Table 2: Relative Path Length (RPL) and Path Success Rate (PSR) for both models across training sample sizes for *Shortest Path*.

| Samples | CogVideoX1.5-5B | | Qwen3-4B-Instruct-2507 | |
|---|---|---|---|---|
| | RPL ↓ | PSR ↑ | RPL ↓ | PSR ↑ |
| 3 | 1.005 | 0.115 | – | – |
| 5 | 1.089 | 0.160 | – | – |
| 10 | 1.060 | 0.245 | – | – |
| 30 | 1.028 | 0.670 | 1.020 | 0.015 |
| 50 | 1.013 | 0.645 | 1.038 | 0.060 |
| 100 | 1.017 | 0.870 | 1.025 | 0.205 |
| 300 | 1.007 | 0.940 | 1.040 | 0.530 |
| 500 | 1.005 | 0.985 | 1.019 | 0.605 |
| 1000 | 1.005 | 0.990 | 1.043 | 0.710 |
| 3000 | 1.000 | 0.990 | 1.026 | 0.795 |
| 5000 | 1.001 | 1.000 | 1.016 | 0.870 |

### 4.2.3 CELLULAR AUTOMATA

We evaluate the capacity of both models to capture complex spatial patterns in cellular automata (CA). Our study spans one-dimensional Elementary Cellular Automata (ECA) Wolfram (1984), a foundational class of binary-state systems, as well as two-dimensional Life-like Cellular Automata, including Conway's Game of Life Gardner (1970), defined by various birth and survival (B/S) rules. Additionally, we consider Langton's ant Langton (1986), a deterministic agent-based system, where the task is to predict the complete grid state after $n$ steps of evolution.

For the 1D ECA experiments, we evaluate four representative rules from each of Wolfram's four complexity classes. We measure task completion as achieving an accuracy above a fixed threshold $\delta = 0.9$. Figure 7 reports the number of training examples required to reach this performance for each rule. Across these rules, both models show broadly similar behavior, with the VDM being better in some cases and worse in others, though overall it remains competitive with the LLM.

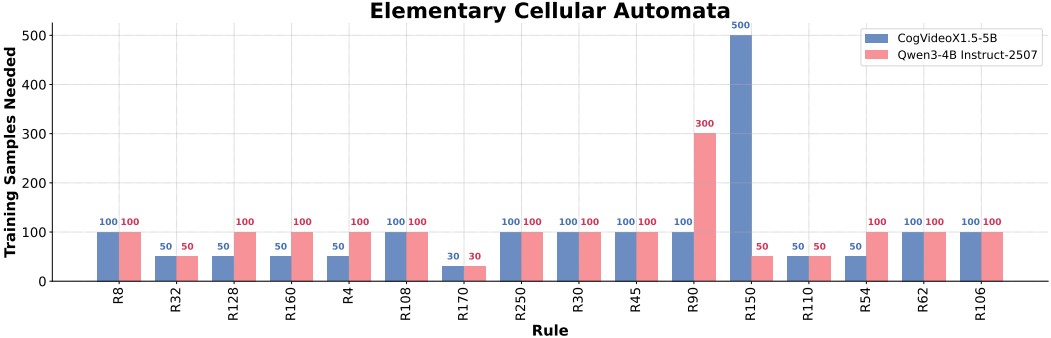

Figure 7: Number of training examples required to achieve $\delta \geq 0.9$ accuracy for selected 1D ECA rules (lower is better).

In two-dimensional settings, clearer differences emerge (see Figures 9, 10). For Life-like cellular automata, the VDM reaches threshold accuracy with far fewer examples, and a similar advantage is observed in Langton's ant. In the case of Langton's ant, the gap grows larger as the number of steps to be predicted increases, indicating that the VDM scales more effectively on tasks that demand long-range spatial planning.

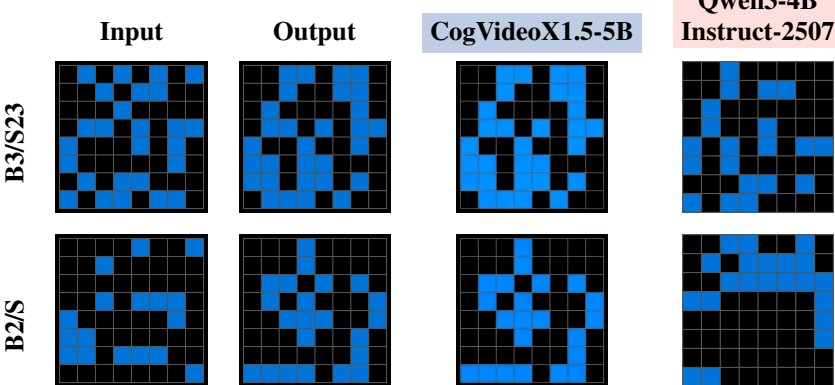

Figure 8: Qualitative examples for Life-like cellular automata with rules *B3/S23* and *B2/S* tasks, after fine-tuning with $n = 30$ samples.

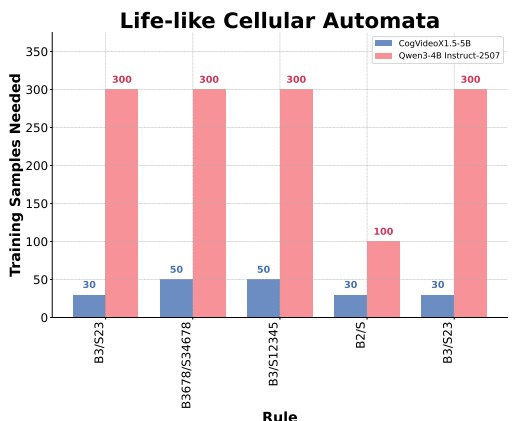

Figure 9: Number of training examples required to achieve $\delta \geq 0.9$ accuracy for selected Life-like cellular automata rules (lower is better).

Figure 10: Accuracy as a function of training set size for **CogVideoX1.5-5B** and **Qwen3-4B-Instruct-2507** on *Langton's Ant* with a prediction horizon of 2,3,5 and 10.

## 5 CONCLUSIONS

Our study shows that VDMs pretrained on spatiotemporal data adapt effectively to structured visual tasks with fewer training examples than comparable LLMs. This demonstrates how modality-aligned pretraining and inductive biases support transfer: VDMs excel in tasks requiring spatial structure and temporal transformation, while LLMs retain strengths in symbol rich domains. Large-scale pretraining on spatiotemporal data with representations aligned to visual structure thus emerges as a promising venue for advancing visual intelligence.

The implications are twofold. For researchers, our benchmarks provide evidence that pretraining pipelines designed around modality-specific structure can unlock new capabilities, offering a path toward more data-efficient models. For practitioners, the inclusion of navigation-style tasks such as mazes and route planning suggests that pretrained VDMs may hold potential for downstream domains like planning, simulation, or robotics. However, validating these capabilities in more realistic, embodied environments remains an important direction for future work.

Overall, these results underline that modality-aligned pretraining plays a central role in advancing visual intelligence.

## REPRODUCIBILITY STATEMENT

We provide detailed hyperparameters and dataset specifications in the Appendix. All synthetic benchmarks are generated with code that we plan to release upon acceptance, together with scripts needed to reproduce the reported results.

## ETHICS STATEMENT

This work investigates pretrained video diffusion models and large language models for visual intelligence and understanding. All datasets used are either publicly available or synthetically generated. No personal, private, or sensitive information is included. We do not anticipate direct ethical risks. We note, however, that advances in generative modeling may be misused for disinformation, and we encourage continued community oversight and safeguards.

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

# APPENDIX

## LIMITATIONS

Our study focuses on grid-based benchmarks such as ARC-AGI, ConceptARC, and synthetic puzzles. This controlled setting provides a systematic way to compare visual and language models, offering a structured interface through which LLMs can express visual understanding. While such benchmarks do not reflect the full diversity of real-world tasks, they are well suited for highlighting the role of modality-aligned pretraining in visual intelligence. Future work should examine whether these insights generalize to more naturalistic and embodied visual environments.

## A EXPERIMENTAL DETAILS

We report here the detailed computational costs and hyperparameter settings used in our experiments. Tables 3 and 4 summarize the GPU hours required across different tasks, while Tables 5 and 6 provide the LoRA fine-tuning configurations for both VDMs and LLMs.

Table 3: GPU hours required for ConceptARC across VDMs and LLMs. Reported hours are wall-clock time and depend on hardware.

| VDM Model (GPU) | Hours | LLM Model (GPU) | Hours |
|---|---|---|---|
| Wan2.1-14B (H100) | 100 | Llama3.1-8B (H100) | 80 |
| LTX-13B (H100) | 95 | Qwen3-8B (2×RTX4090) | 100 |
| CogVideoX1.5-5B (RTX4090) | 130 | Qwen3-4B-Instruct-2057 (RTX4090) | 135 |
| LTX-2B (H100) | 40 | | |

Table 4: GPU hours required for ARC-AGI and Structured Visual Tasks. Reported hours are wall-clock time and depend on hardware.

| ARCAGI Model (GPU) | Hours | Structured Task Model (GPU) | Hours |
|---|---|---|---|
| CogVideoX1.5-5B (RTX4090) | 450 | CogVideoX1.5-5B (RTX4090) | 1650 |
| Qwen3-4B-Instruct-2057 (RTX4090) | 475 | Qwen3-4B-Instruct-2057 (RTX4090) | 2000 |

To ensure reproducibility, we also include the fine-tuning hyperparameters for each model. The following two tables detail the LoRA, training, and optimizer configurations used for VDMs (Table 5) and LLMs (Table 6).

Table 5: LoRA finetuning configuration for VDM experiments.

| Parameter | LTX-13B | LTX-2B | CogVideoX1.5-5B | Wan2.1-14B |
|---|---|---|---|---|
| *LoRA Configuration* | | | | |
| Rank | 64 | 64 | 64 | 64 |
| Alpha | 64 | 64 | 32 | 32 |
| Target modules | to_q, to_k, to_v, to_out.0, ff.net.0.proj, ff.net.2 | to_q, to_k, to_v, to_out.0, ff.net.0.proj, ff.net.2 | QKVO | – |
| *Training Configuration* | | | | |
| Seed | 42 | 42 | 42 | 42 |
| Batch size | 2 | 4 | 2 | 1 |
| Gradient accumulation steps | 2 | 1 | 1 | 1 |
| *Optimizer Configuration* | | | | |
| Optimizer | AdamW | AdamW | AdamW | AdamW |
| Learning rate | 2e-4 | 2e-4 | 1e-4 | 1e-4 |
| Scheduler | Linear | Linear | Constant | Constant |
| Max grad norm | 1.0 | 1.0 | 1.0 | 0.05 |

*Note.* LoRA ranks differ slightly across model families (VDMs use rank 64, whereas LLMs use rank 32). We verified that performance is largely insensitive to this setting: Qwen3 models with rank 64 perform comparably to rank 32, and CogVideoX1.5-5B models with rank 32 match the reported rank 64 results. In both cases, we report the configuration that yielded stronger results in our initial trials. All reported results in the paper correspond to the configurations shown in the tables.

## B TASK DETAILS

For completeness, we provide additional explanations of the tasks considered in our evaluation. Each subsection introduces a task family and highlights the key rules and objectives, we further provide examples on how the task is encoded into image and text.

Table 6: LoRA finetuning configuration for LLMs used.

| Parameter | Qwen3-4B-Instruct-2507 | Qwen3-8B | LLaMA-3.1-8B |
|---|---|---|---|
| *LoRA Configuration* | | | |
| Rank | 32 | 32 | 32 |
| Alpha | 32 | 32 | 64 |
| Dropout | 0 | 0 | 0.05 |
| Target modules | q_proj, k_proj, v_proj, o_proj, gate_proj, up_proj, down_proj | q_proj, k_proj, v_proj, o_proj, gate_proj, up_proj, down_proj | q_proj, k_proj, v_proj, o_proj, gate_proj, up_proj, down_proj, lm_head |
| *Model Setup* | | | |
| Max sequence length | 8192 | 8192 | 4096 |
| Random seed | 3407 | 3407 | 3407 |
| *Training Configuration* | | | |
| Batch size per device | 2 | 1 | 1 |
| Effective batch size | 8 | 8 | 8 |
| Gradient accumulation steps | 4 | 8 | 8 |
| Learning rate | 2e-4 | 2e-4 | 2e-4 |
| Scheduler | Linear | Linear | Linear |
| Warmup steps | 5 | 5 | 5 |
| Weight decay | 0.01 | 0.01 | 0.01 |
| *Generation Configuration* | | | |
| Max new tokens | 4096 | 4096 | 4096 |
| Temperature | 0.7 | 0.7 | 0.7 |
| Top-$p$ | 0.8 | 0.8 | 0.8 |
| Top-$k$ | 20 | 20 | 20 |

## B.1 VISUAL GAMES

### B.1.1 HITORI 5x5

**Objective:** Eliminate cells so that each number appears at most once per row and column.

**Rules:**

1. A number must not be repeated in any row or column.

2. Shaded cells cannot be orthogonally adjacent.

3. All unshaded cells must form a single connected component.

We add an example of the task in Figure 7.

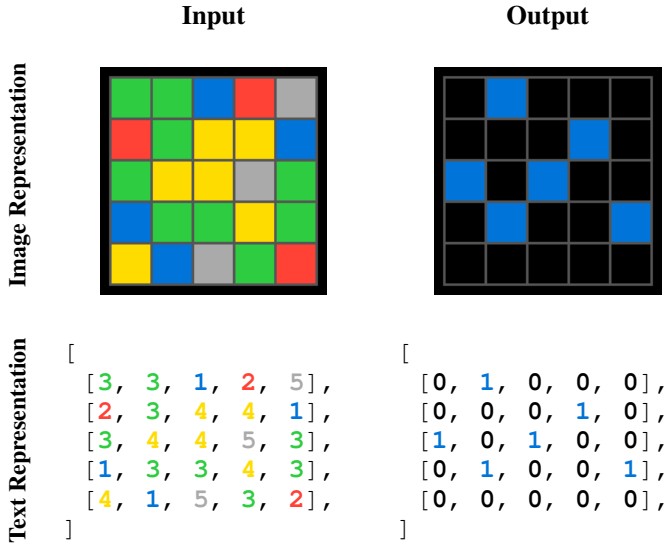

Table 7: Example input-output pair for task *Hitori*.

### B.1.2   SUDOKU

**Objective:** Fill the grid so that all constraints are satisfied.

**Rules:**

1. Each row must contain all required digits without repetition.
2. Each column must contain all required digits without repetition.
3. Each subgrid must contain all required digits without repetition.

We evaluate two variants: *Mini Sudoku* (4x4 with 2x2 subgrids, see Figure 11) and *Sudoku* (9x9 with 3x3 subgrids, see Figure 12).

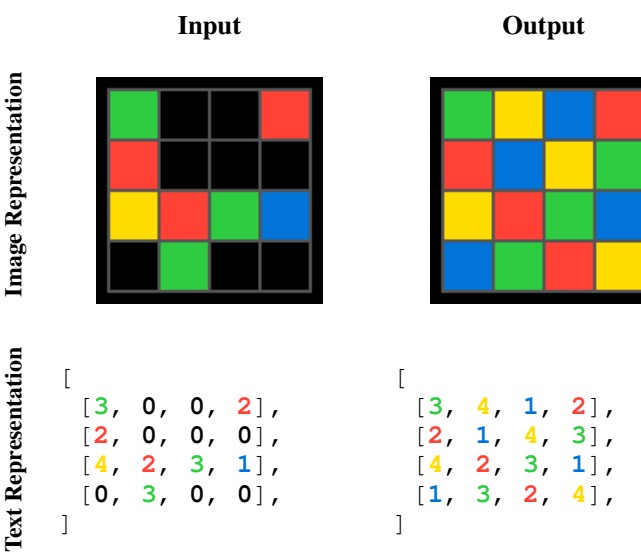

Figure 11: Example input-output pair for task *Sudoku Mini*.

### B.1.3   CONNECT 4

**Objective:** Place tokens to align four in a row.

**Rules:**

1. Players alternate dropping tokens into one of the seven columns.
2. A token occupies the lowest available cell in the chosen column.
3. A player wins by forming a horizontal, vertical, or diagonal line of four tokens.

We restrict evaluation to single-move winning scenarios, see Figure 13.

### B.1.4   CHESS MATE-IN-1

**Objective:** Deliver checkmate in a single move. **Rules:**

1. All standard chess movement rules apply.
2. A move is correct only if it results in an immediate checkmate of the opposing king.

To ensure the task is well defined, we filter scenarios so that they always correspond to white moves. The original dataset is extracted from quantum24 (2023), and an illustrative example is shown in Figure 14.

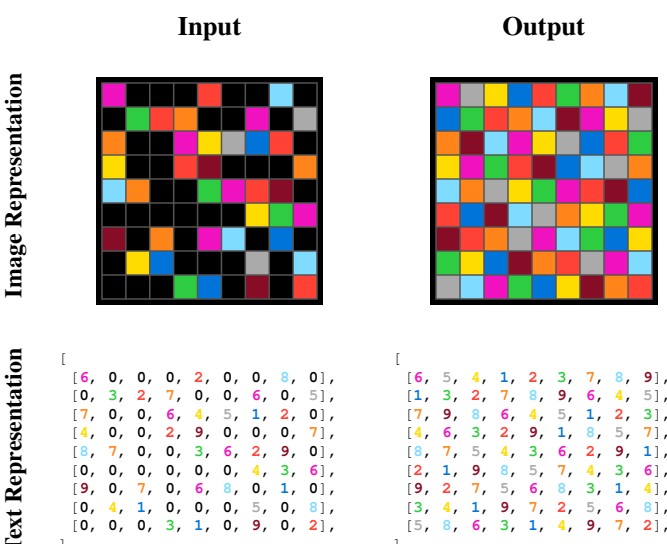

Figure 12: Example input-output pair for task *Sudoku*.

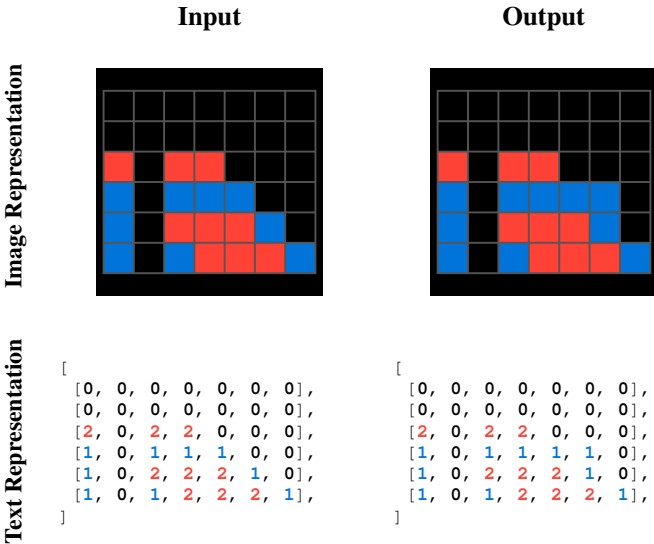

Figure 13: Example input-output pair for task *Connect4*.

## B.2 ROUTE PLANNING

We evaluate route planning in two-dimensional grid environments. The objective across tasks is to construct valid paths that connect designated start and goal locations under different structural constraints. We consider two tasks: *Maze* and *Shortest Path*.

### B.2.1 MAZE

**Objective:** Navigate from the start cell to the goal cell through a grid containing blocked and open positions.

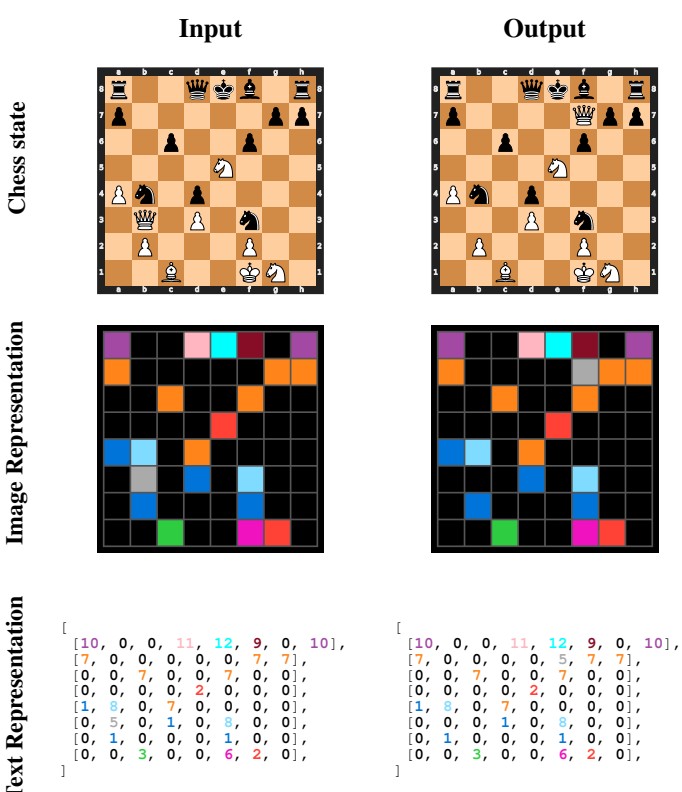

Figure 14: Example input-output pair for task *Chess Mate in 1*.

**Rules:**

1. The agent starts at the top-left cell and must reach the bottom-right cell.

2. Movement is allowed only through open cells.

3. Allowed moves are up, down, left, and right (no diagonal moves).

4. A valid solution is a continuous sequence of moves from start to goal.

We evaluate two scenarios:

- **Base Maze:** Training and evaluation on $21 \times 21$ grids.
- **Maze Generalization:** Training on smaller $13 \times 13$ grids and testing on larger $21 \times 21$ grids.

We illustrate a sample $21 \times 21$ maze in Figure 16, which serves as training and evaluation data in the *Base Maze* setting and as evaluation data in the *Maze Generalization* setting. Figure 15 shows a sample $13 \times 13$ maze, which is used as training data in the *Maze Generalization* setting.

### B.2.2 SHORTEST PATH

**Objective:** Connect two arbitrary points with the shortest possible route.

**Rules:**

1. Start and goal cells are specified anywhere on the grid.

2. Movement is allowed only through open cells.

3. Allowed moves are up, down, left, and right (no diagonal moves).

**Input** **Output**

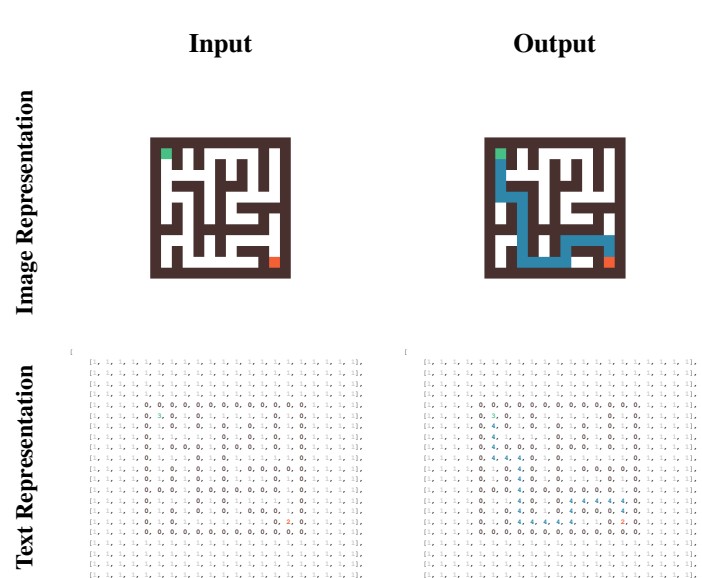

Figure 15: Example input-output pair for task *Maze Small*.

**Input** **Output**

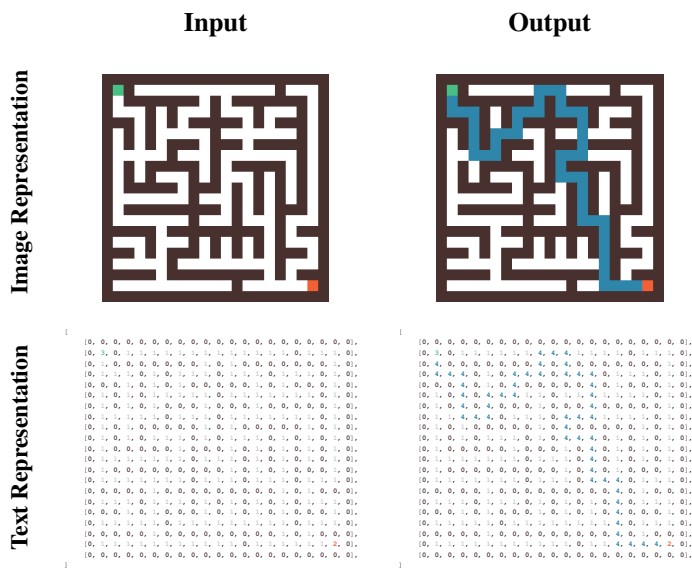

Figure 16: Example input-output pair for task *Maze*.

4. A valid solution is a continuous path from start to goal with minimal length among all possible paths.

We provide an example in Figure 17.

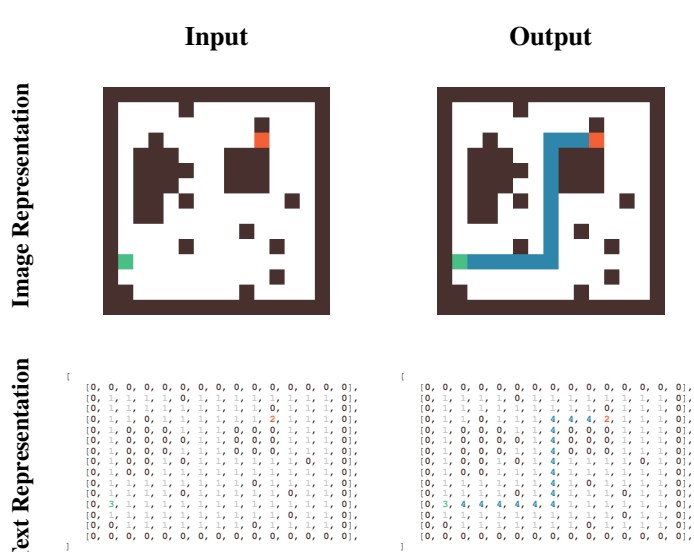

Figure 17: Example input-output pair for task *Shortest Path*.

Table 8: Representative Elementary Cellular Automata rules by Wolfram class.

| Class | Rules |
|-------|-------|
| Class 1 | 8, 32, 128, 160 |
| Class 2 | 4, 108, 170, 250 |
| Class 3 | 30, 45, 90, 150 |
| Class 4 | 110, 54, 62, 106 |

## B.3 CELLULAR AUTOMATA

### B.3.1 ELEMENTARY CELLULAR AUTOMATA (ECA)

Elementary Cellular Automata (ECA) are one-dimensional binary-state automata defined on a line of cells. Each cell $c_i^t \in \{0, 1\}$ at time $t$ updates based on itself and its two neighbors:

$$c_i^{t+1} = f(c_{i-1}^t, c_i^t, c_{i+1}^t),$$

where $f$ is specified by a rule number between 0 and 255.

For example, Rule 110 is encoded by the binary string `01101110`, which maps the eight possible neighborhoods $(c_{i-1}^t, c_i^t, c_{i+1}^t)$ to the next state:

| Neighborhood | 111 | 110 | 101 | 100 | 011 | 010 | 001 | 000 |
|--------------|-----|-----|-----|-----|-----|-----|-----|-----|
| Next state | 0 | 1 | 1 | 0 | 1 | 1 | 1 | 0 |

We evaluate four representative rules from each of Wolfram's classes Wolfram (1984), summarized in Table 8.

Rule 110 is well known for its complex localized structures and universality Cook (2004). We show an example in Figure 18.

### B.3.2 LIFE-LIKE CELLULAR AUTOMATA

Life-like CA generalize Conway's Game of Life Gardner (1970), using binary cells on a two-dimensional grid. Each cell updates according to the number of live neighbors in the Moore neigh-

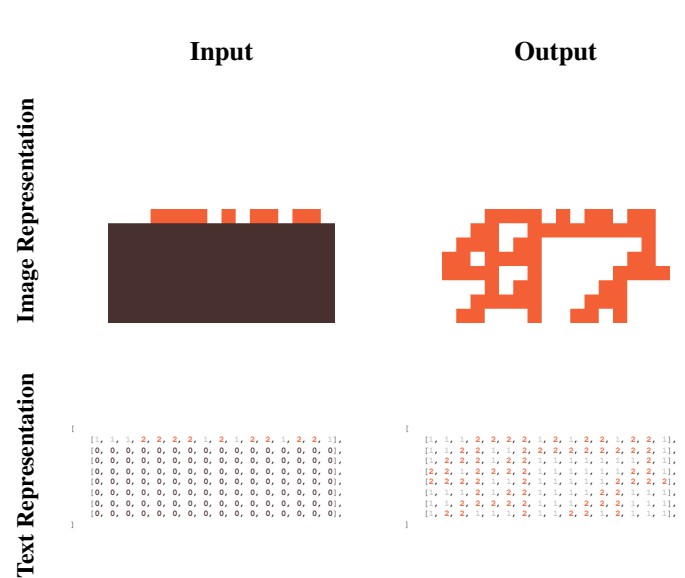

Figure 18: Example input-output pair for task *Langton ant step 2*.

borhood (eight adjacent cells). In standard Game of Life ($B3/S23$):

$$c_{i,j}^{t+1} = \begin{cases} 1 & \text{if cell is dead and has exactly 3 live neighbors (birth),} \\ 1 & \text{if cell is alive and has 2 or 3 live neighbors (survival),} \\ 0 & \text{otherwise (death).} \end{cases}$$

We consider several well-known Life-like variants. These rules, summarized in Table 9, capture diverse behaviors ranging from explosive growth to symmetry under inversion. We shown an example in Figure 19 of the basic Game of Life.

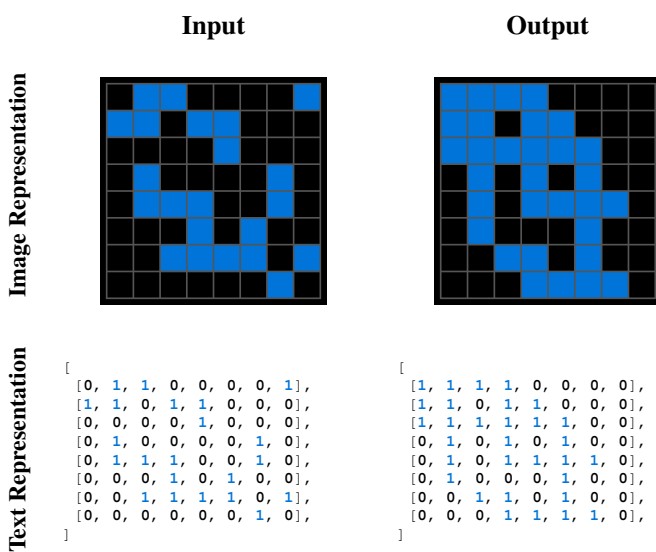

Figure 19: Example input-output pair for task *Game of Life step 1*.

Table 9: Life-like cellular automata variants evaluated.

| Name | Rule (B/S) | Description |
|------|-----------|-------------|
| Day & Night | B3678/S34678 | Symmetric under inversion; complex dynamics |
| Maze | B3/S12345 | Generates labyrinth-like, maze-like growth |
| Seeds | B2/S$\varnothing$ | All live cells die each step; explosive expansion |
| Life | B3/S2 | Sparse survival; promotes small, mobile clusters |

### B.3.3 LANGTON'S ANT

Langton's ant Langton (1986) is an agent-based CA where a single agent moves on a binary grid. At each step:
$$(x, y), d, g(x, y) \rightarrow (x', y'), d', g'(x, y),$$
where $(x, y)$ is the current cell, $d$ is direction, and $g(x, y) \in \{0, 1\}$ is the cell state.

1. If $g(x, y) = 0$, turn right; if $g(x, y) = 1$, turn left.

2. Flip the cell color: $g'(x, y) = 1 - g(x, y)$.

3. Move forward one step.

After many steps, chaotic behavior gives way to a repeating "highway" structure. To make the task predictable, **we always start with the ant facing on the same initial direction and being on top of a 0 cell.** For an example see Figure 20

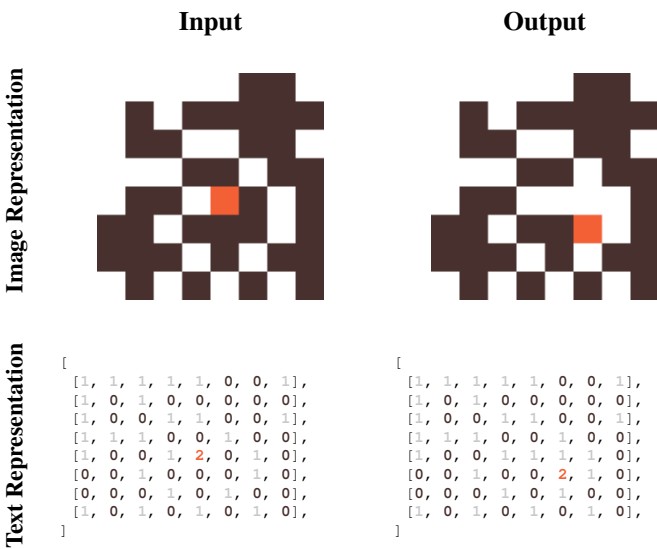

Figure 20: Example input-output pair for task *Langton ant step 2*.

## C ADDITIONAL QUALITATIVE RESULTS

### C.1 ARC-AGI

To further illustrate the complementary strengths of VDMs and LLMs, we include qualitative examples of ARC-AGI tasks. In some cases, the LLM enables it to find the correct solution, while the VDM fails. Examples of this behavior is shown in Figure 23.

In contrast, there are tasks where both models succeed, suggesting that the underlying structure can be captured through either symbolic reasoning or visual pattern learning. One such case is given in Figure 24.

Finally, we highlight situations where only the VDM solves the task correctly (Figures 21 and 22). These examples emphasize how visual inductive biases allow the VDM to generalize in settings where symbolic reasoning alone appears insufficient.

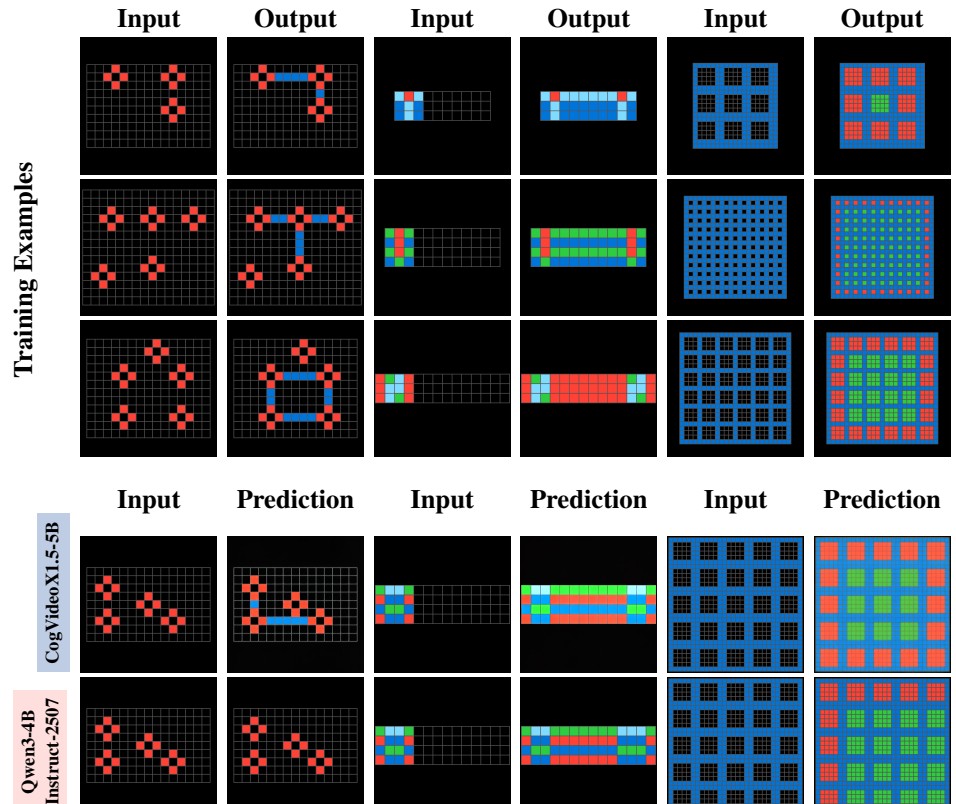

Figure 21: Qualitative results on ARC-AGI for problems *60a26a3e*, *62b74c02*, *8a371977*.

## C.2 STRUCTURED VISUAL TASKS

We include additional qualitative examples from structured visual tasks such as mazes, route planning, and cellular automata, complementing the quantitative results in the main text.

## D ADDITIONAL RESULTS

## E ARC FAMILY

Here, we include the comparison table for ConceptARC, by including finetuned LLMs (Qwen3-4B-Instruct, Qwen3-8B, LLama3.1-8B-Instruct) and GPT-4 [IC][2] Moskvichev et al. (2023), as well as VDMs (CogVideoX1.5-5B, Wan2.1-14B, LTX-2B/13B). These additional results provide broader context and help reinforce the trends observed in the main text. See Table 10.

The relatively lower performance of LTX compared to other VDMs may stem from its aggressive VAE compression, which can discard structural information important for ConceptARC. This reflects a design tradeoff of the LTX models, aimed at enabling much faster video generation HaCohen et al. (2025).

---

[2]Added for reference with commercial models, this case is directly IC and not our finetune approach.

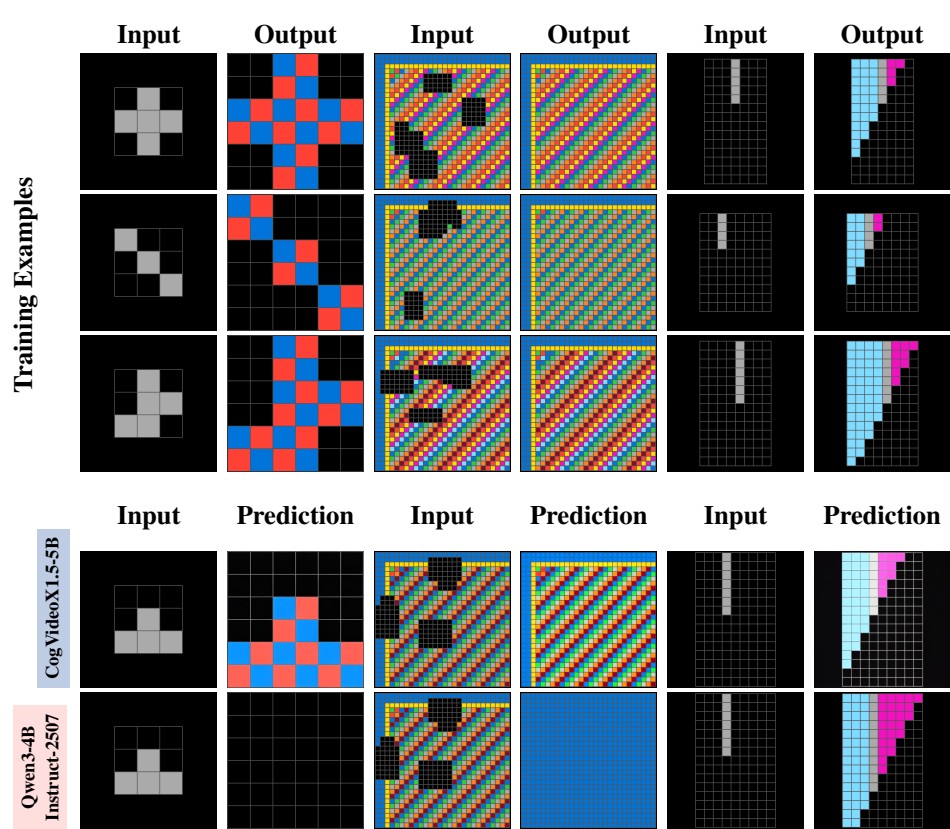

Figure 22: Qualitative results on ARC-AGI for problems *2072aba6*, *4aab4007*, *5207a7b5*.

Table 10: Concept-wise overall accuracy across models. Best values are highlighted for **VDMs** or **LLMs** .

| Concept | LTX-13B | LTX-2B | Wan2.1-14B | CogVideoX1.5-5B | Qwen3-4B Instruct-2507 | Qwen3-8B | Llama3.1-8B | GPT-4 [IC] |
|---|---|---|---|---|---|---|---|---|
| AboveBelow | 0.30 | 0.17 | 0.37 | **0.40** | **0.40** | **0.40** | 0.17 | 0.23 |
| TopBottom2D | 0.23 | 0.17 | **0.63** | 0.37 | 0.50 | 0.50 | 0.37 | 0.23 |
| TopBottom3D | 0.27 | 0.17 | **0.47** | 0.33 | 0.13 | 0.20 | 0.17 | 0.20 |
| HorizontalVertical | 0.13 | 0.20 | **0.53** | 0.47 | 0.43 | 0.47 | 0.33 | 0.27 |
| Center | 0.33 | 0.30 | **0.57** | 0.37 | 0.20 | 0.20 | 0.13 | 0.33 |
| FilledNotFilled | 0.30 | 0.27 | **0.50** | 0.37 | 0.27 | 0.23 | 0.20 | 0.17 |
| CompleteShape | 0.20 | 0.10 | **0.40** | 0.37 | 0.23 | 0.30 | 0.13 | 0.23 |
| InsideOutside | 0.27 | 0.27 | **0.37** | 0.33 | 0.13 | 0.20 | 0.13 | 0.10 |
| ExtractObjects | 0.07 | 0.07 | **0.23** | 0.07 | 0.10 | 0.10 | 0.03 | 0.03 |
| Count | 0.40 | 0.43 | **0.83** | 0.57 | 0.13 | 0.13 | 0.17 | 0.13 |
| SameDifferent | 0.23 | 0.23 | 0.33 | **0.37** | 0.27 | 0.23 | 0.27 | 0.17 |
| Order | 0.03 | 0.03 | 0.00 | 0.07 | **0.27** | **0.27** | 0.10 | **0.27** |
| MoveToBoundary | 0.17 | 0.00 | 0.13 | 0.17 | **0.23** | 0.10 | 0.17 | 0.20 |
| ExtendToBoundary | 0.20 | 0.23 | **0.50** | 0.40 | 0.13 | 0.17 | 0.10 | 0.07 |
| Copy | 0.20 | 0.03 | 0.17 | 0.13 | 0.17 | 0.10 | 0.10 | **0.23** |
| CleanUp | 0.43 | 0.40 | **0.60** | 0.53 | 0.27 | 0.30 | 0.27 | 0.20 |
| **Average Accuracy** | 0.24 | 0.19 | **0.41** | 0.33 | 0.24 | 0.24 | 0.18 | 0.19 |

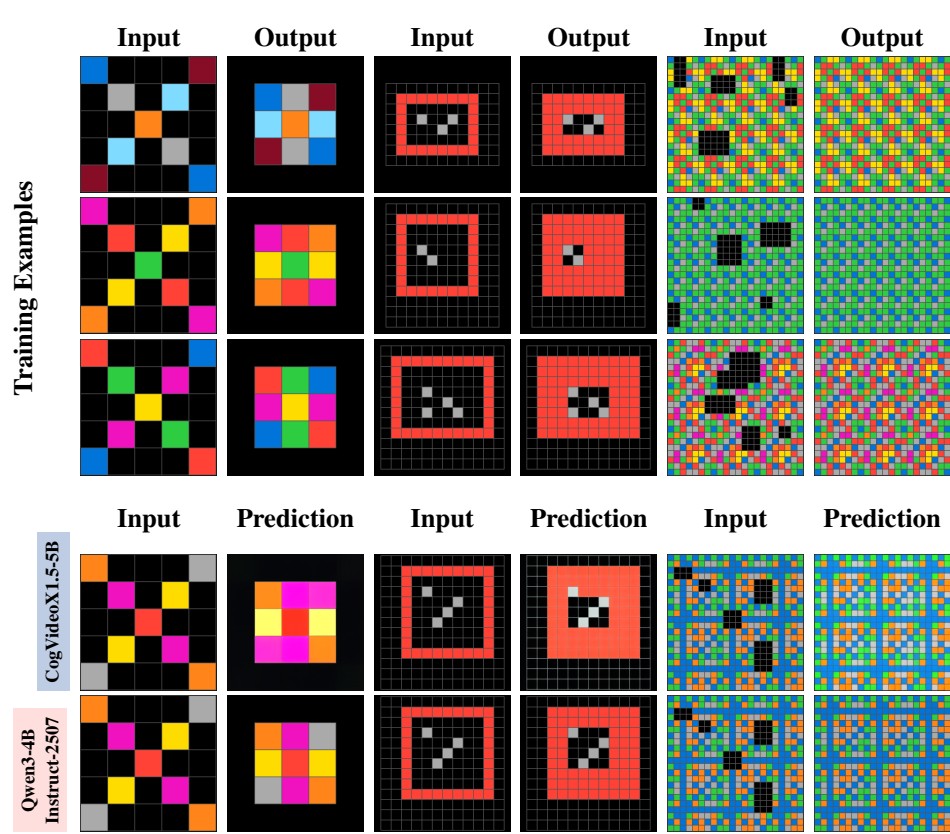

Figure 23: Qualitative results on ARC-AGI for problems *ca8de6ea*, *d37a1ef5*, *e95e3d8e*.

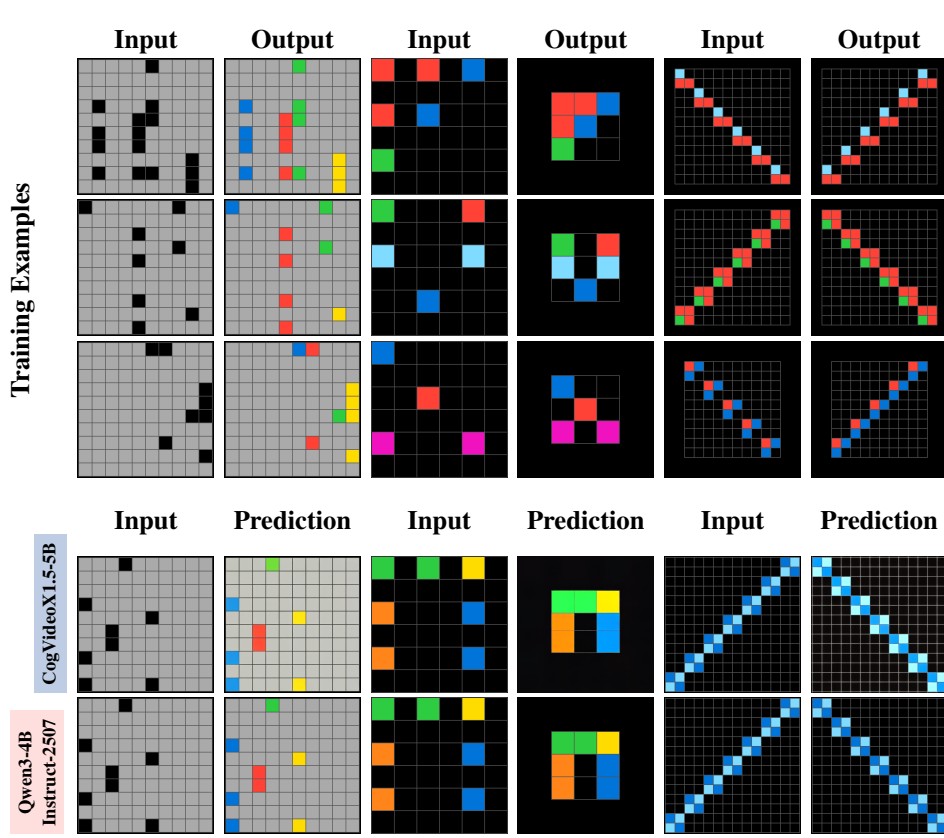

Figure 24: Qualitative results on ARC-AGI for problems *575b1a71*, *68b67ca3*, *8ee62060*.

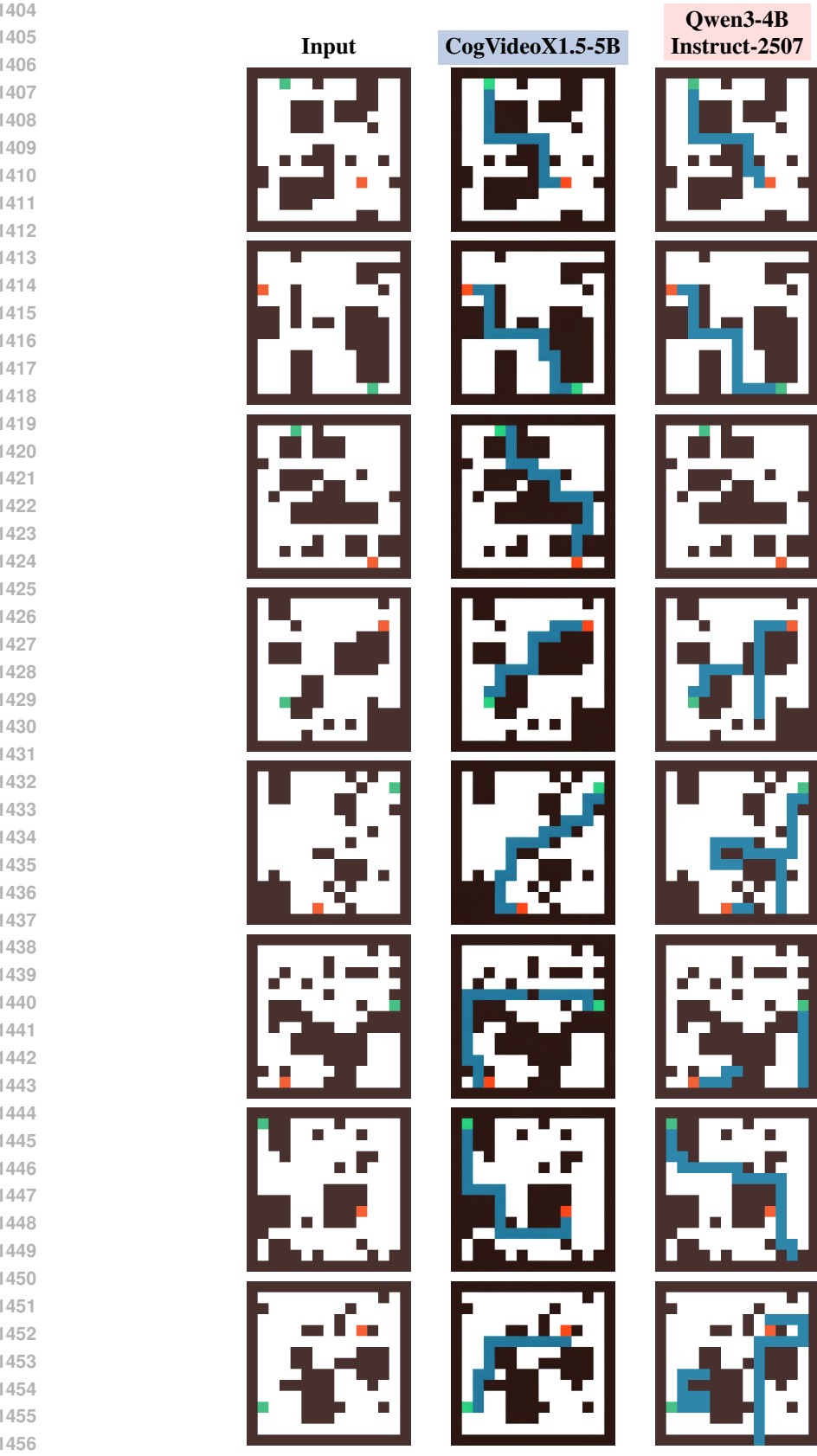

Figure 25: Representative examples for the *Shortest Path* task, showing ground truth inputs (left) and model predictions (center and right) after finetuning with $n = 300$ samples.

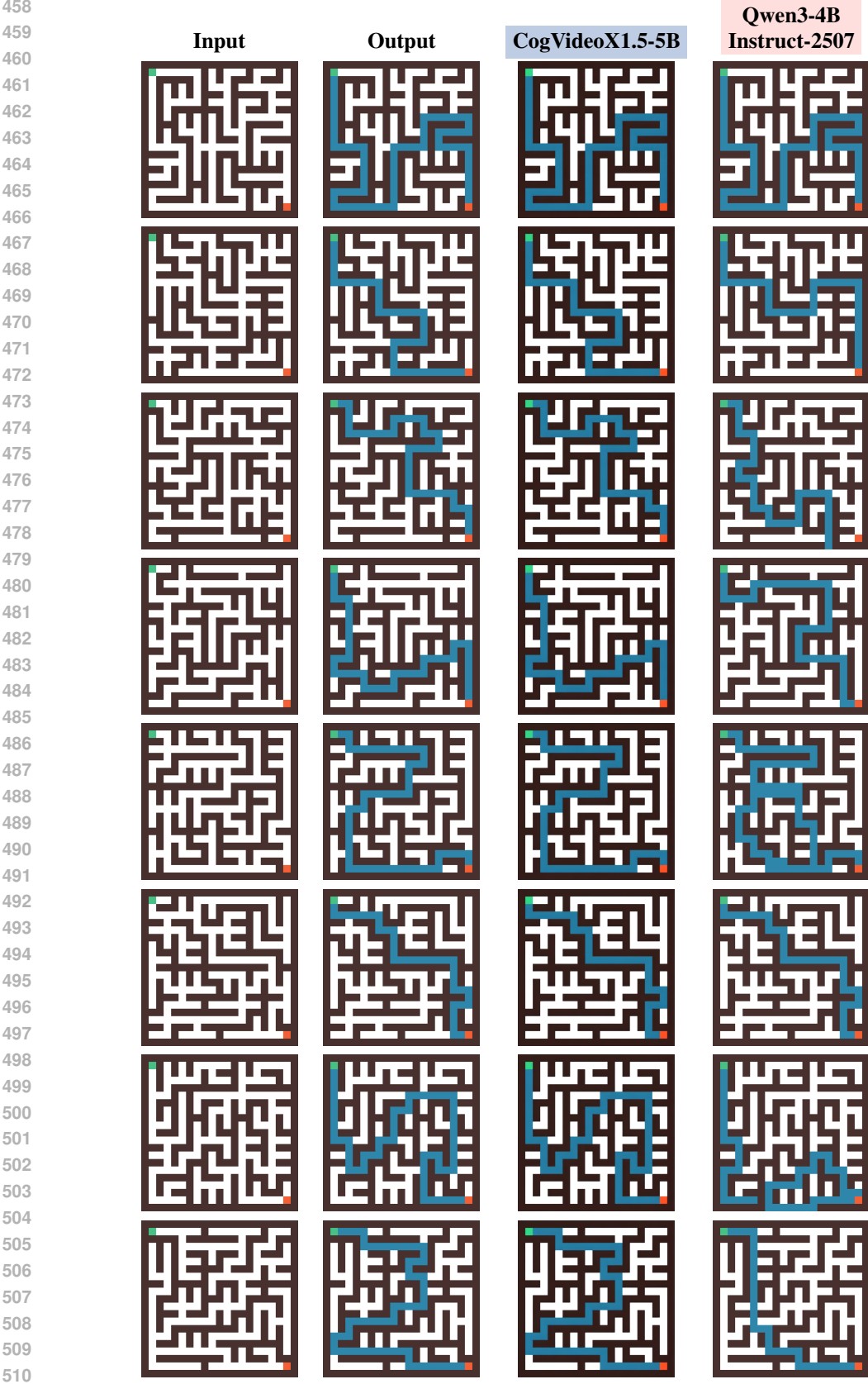

Figure 26: Additional qualitative examples for the *Maze* task, showing inputs, ground truth outputs, and model predictions after finetuning with $n = 300$ samples.

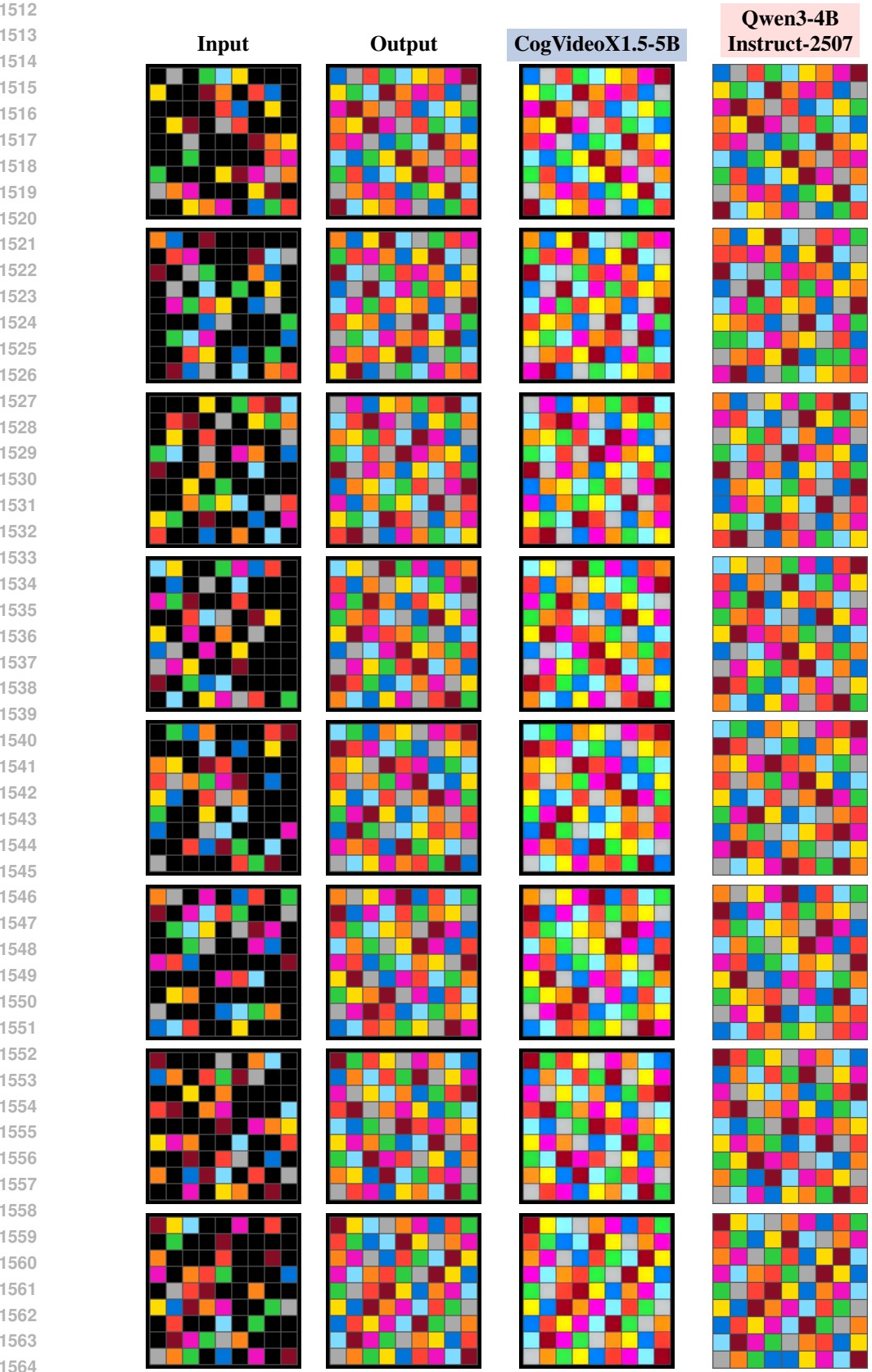

Figure 27: Additional qualitative examples for the *Sudoku* task, showing inputs, ground truth outputs, and model predictions after finetuning with $n = 1000$ samples.

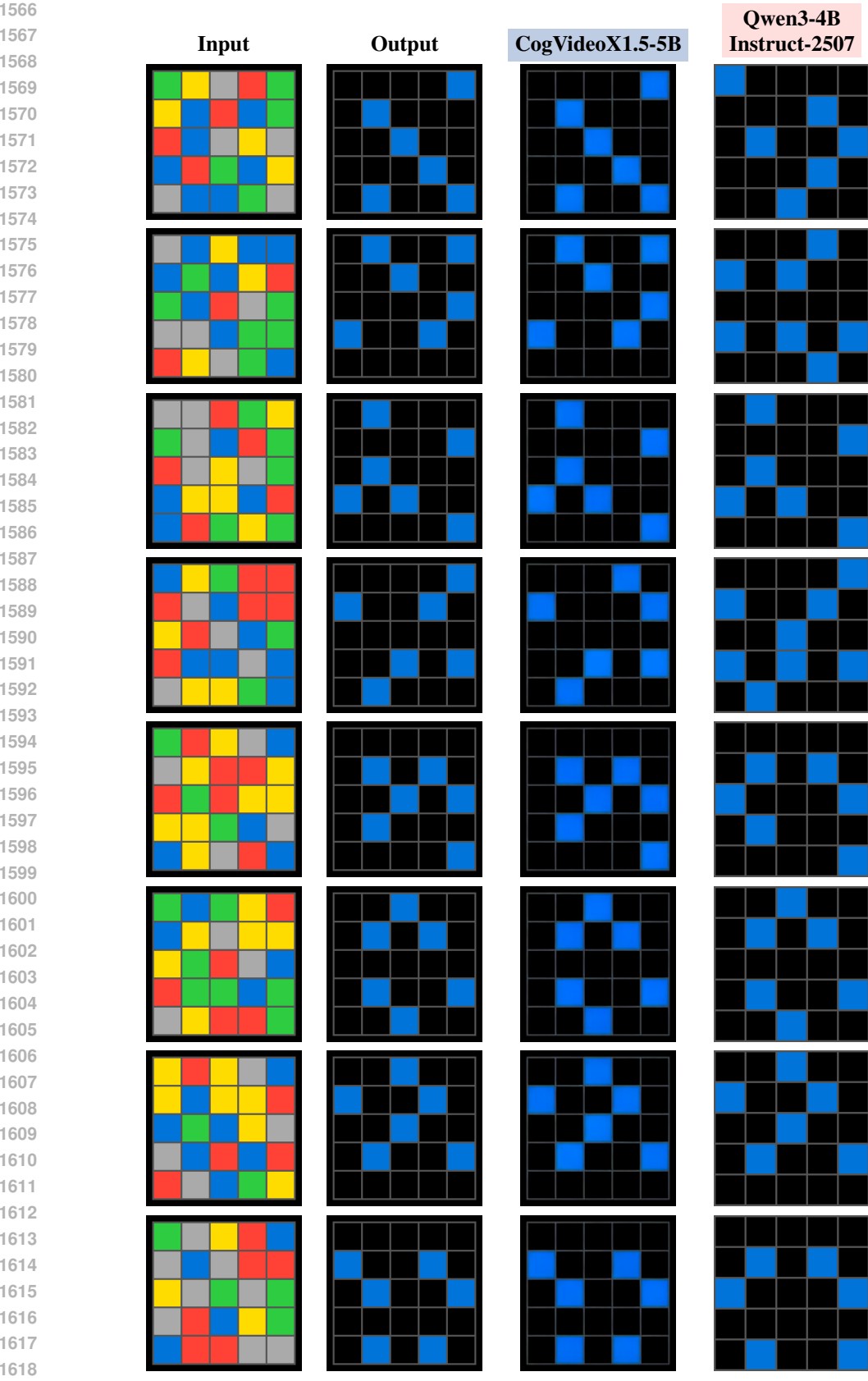

Figure 28: Additional qualitative examples for the *Hitori* task, showing inputs, ground truth outputs, and model predictions after finetuning with $n = 100$ samples.

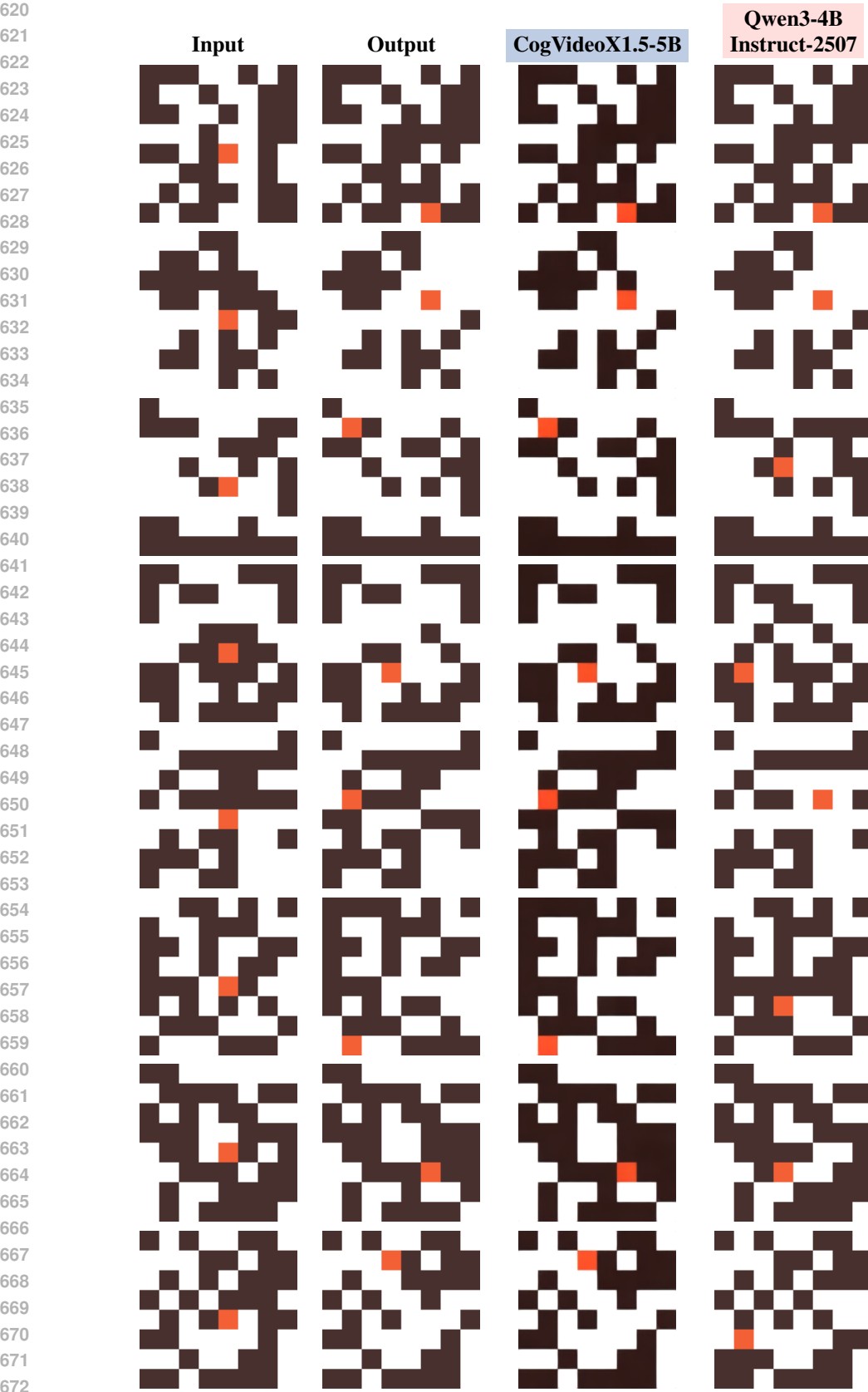

Figure 29: Additional qualitative examples for the *Langton Ant* (horizon 10) task, showing inputs, ground truth outputs, and model predictions after finetuning with $n = 1000$ samples.

## E.1 Pitfalls of Vision Language Models

Vision–Language Models (VLMs) promise to bridge the gap between visual perception and language by training on vast datasets of paired images and text. In principle, this multimodal pretraining should enable these models to solve visually grounded tasks more effectively than language-only models. To test whether this promise holds in practice, we evaluate a representative VLM, Gemma-4B Gemma Team (2025), on a structured visual task: *Sudoku*.

We fine-tune the same model with $n = 1000$ samples under three configurations: **text-only**, **image-only**, and **combined image–text**; keeping all other settings fixed. The results in Table 11 reveal a striking limitation: adding image input offers no measurable improvement, and the **image-only** variant performs worse than a trivial baseline. This suggests that the model is unable to extract meaningful information from visual inputs, even when explicitly trained to do so.

Table 11: Relative Accuracy and Accuracy on *Sudoku*.

| Model | Relative Accuracy | Accuracy |
|---|---|---|
| **Text-only** | 0.79 | 0.06 |
| **Combined image–text** | 0.78 | 0.06 |
| **Image-only** | 0.12 | 0.00 |

To investigate why, we train the **image-only** model on a simplified task: reconstructing the textual grid representation of its own image input rather than predicting a Sudoku solution. With small training sets ($n = 3, 5, 10$), the model fails to interpret the images and instead memorizes training samples, reproducing them verbatim regardless of input (Table 12). The model learns little about the underlying structure of the visual input.

Table 12: Distribution of outputs on the test set exactly matching training samples for different training set sizes.

| Training Set Size | Sample | Proportion | Total Proportion |
|---|---|---|---|
| 3 | Sample 1 | 0.385 | |
| | Sample 2 | 0.010 | **1.00** |
| | Sample 3 | 0.605 | |
| 5 | Sample 1 | 0.490 | |
| | Sample 2 | 0.030 | |
| | Sample 3 | 0.335 | **0.99** |
| | Sample 4 | 0.135 | |
| 10 | Sample 1 | 0.100 | |
| | Sample 2 | 0.010 | |
| | Sample 3 | 0.030 | |
| | Sample 4 | 0.005 | |
| | Sample 5 | 0.015 | **0.96** |
| | Sample 6 | 0.170 | |
| | Sample 7 | 0.615 | |
| | Sample 8 | 0.010 | |

This experiment exposes a deeper issue: despite their multimodal pretraining, current VLMs struggle to extract structured information from images Jing et al. (2025); Sim et al. (2025). They appear to rely primarily on semantics and basic pattern recognition rather than true visual understanding. Furthermore, VLMs inherit many of the limitations of LLMs, such as reliance on text-based outputs, without gaining meaningful visual understanding ability.

Because VLMs provide no measurable advantage over language-only models for these structured visual tasks, we focus on LLMs as the primary baseline. LLMs already demonstrate strong capabilities in structured prediction and symbolic manipulation, making them a fair and informative comparison point for VDMs. This framing keeps the evaluation focused on model families that offer complementary strengths.

## F  Results - Full Tables

We provide the complete set of experimental results, which constitute the underlying data for the figures reported in the main paper.

Table 13: Comparison of CogVideoX1.5-5B and Qwen3-4B-Instruct-2507 accuracy on structured games. Missing values are shown as −.

| n | CogVideoX1.5-5B | | | | | Qwen3-4B-Instruct-2507 | | | | |
|---|---|---|---|---|---|---|---|---|---|---|
| | Chess-Mate-in-1 | Connect 4 | Hitori 5x5 | Sudoku Mini | Sudoku | Chess-Mate-in-1 | Connect 4 | Hitori 5x5 | Sudoku Mini | Sudoku |
| 3 | 0.00 | 0.44 | 0.01 | 0.22 | 0.00 | 0.00 | 0.03 | 0.00 | 0.18 | − |
| 5 | 0.00 | 0.62 | 0.02 | 0.36 | 0.00 | 0.02 | 0.05 | 0.00 | 0.22 | − |
| 10 | 0.00 | 0.74 | 0.62 | 0.65 | 0.00 | 0.04 | 0.08 | 0.02 | 0.48 | − |
| 30 | 0.02 | 0.78 | 0.72 | 0.78 | 0.20 | 0.13 | 0.38 | 0.02 | 0.64 | 0.00 |
| 50 | 0.04 | 0.80 | 0.84 | 0.90 | 0.34 | 0.15 | 0.38 | 0.10 | 0.68 | 0.00 |
| 100 | 0.08 | 0.85 | 0.92 | 0.91 | 0.60 | 0.24 | 0.69 | 0.28 | 0.78 | 0.01 |
| 300 | 0.14 | 0.84 | 0.94 | 0.90 | 0.55 | 0.38 | 0.71 | 0.57 | 0.80 | 0.01 |
| 500 | 0.20 | 0.89 | 0.94 | 0.94 | 0.60 | 0.44 | 0.69 | 0.64 | 0.86 | 0.06 |
| 1000 | 0.22 | 0.90 | 0.96 | 0.91 | 0.79 | 0.56 | 0.76 | 0.86 | 0.90 | 0.14 |
| 3000 | − | 0.92 | 0.98 | 0.95 | 0.86 | − | 0.78 | 0.94 | 0.92 | 0.32 |
| 5000 | − | 0.90 | 0.99 | 0.96 | 0.86 | − | 0.82 | 0.96 | 0.96 | 0.55 |

Table 14: Comparison of CogVideoX1.5-5B and Qwen3-4B-Instruct-2507 accuracy on Life-Like Cellular Automata variants. Missing values are shown as −.

| n | CogVideoX1.5-5B | | | | | Qwen3-4B-Instruct-2507 | | | | |
|---|---|---|---|---|---|---|---|---|---|---|
| | Life_B3S2 | DayAndNight | Maze | Seeds | Game of Life | Life_B3S2 | DayAndNight | Maze | Seeds | Game Of Life |
| 10 | 0.00 | 0.00 | 0.00 | 0.00 | 0.00 | − | − | − | − | − |
| 30 | 1.00 | 0.81 | 0.87 | 1.00 | 0.96 | − | 0.63 | 0.81 | 0.75 | 0.63 |
| 50 | 1.00 | 0.95 | 0.91 | 1.00 | 0.97 | − | 0.64 | 0.80 | 0.78 | 0.64 |
| 100 | 1.00 | 1.00 | 0.96 | 1.00 | 1.00 | 0.61 | 0.70 | 0.87 | 0.63 | 0.73 |
| 300 | − | − | − | − | − | 1.00 | 1.00 | 1.00 | 1.00 | 1.00 |
| 500 | − | − | − | − | − | − | 1.00 | 1.00 | 1.00 | 1.00 |

Table 15: Comparison of CogVideoX1.5-5B and Qwen3-4B-Instruct-2507 accuracy on Langton's Ant with respect to number of steps into the future. Missing values are shown as −.

| n | CogVideoX1.5-5B | | | | Qwen3-4B-Instruct-2507 | | | |
|---|---|---|---|---|---|---|---|---|
| | Step 2 | Step 3 | Step 5 | Step 10 | Step 2 | Step 3 | Step 5 | Step 10 |
| 3 | 0.18 | 0.03 | 0.03 | − | 0.32 | 0.03 | − | − |
| 5 | 0.23 | 0.07 | 0.04 | 0.00 | 0.21 | 0.04 | − | − |
| 10 | 0.67 | 0.29 | 0.06 | 0.01 | 0.51 | 0.19 | − | − |
| 30 | 1.00 | 0.76 | 0.25 | 0.01 | 0.79 | 0.46 | 0.06 | 0.00 |
| 50 | 1.00 | 0.99 | 0.41 | 0.01 | 0.950 | 0.58 | 0.14 | 0.010 |
| 100 | 1.00 | 1.000 | 0.88 | 0.08 | 0.99 | 0.910 | 0.39 | 0.01 |
| 300 | − | − | 1.00 | 0.42 | 1.00 | 1.00 | 0.98 | 0.12 |
| 500 | − | − | 1.00 | 0.83 | 1.00 | 1.00 | 1.00 | 0.21 |
| 1000 | − | − | 1.00 | 0.98 | 1.00 | 1.00 | 1.00 | 0.47 |
| 3000 | − | − | − | 0.99 | − | − | − | 0.71 |
| 5000 | − | − | − | − | − | − | − | 0.93 |

## G  Exploring Generalization of I2I-Tuned VDMs

While the main text emphasizes grid-structured visual prediction tasks, our framework extends naturally to a broad range of image-to-image problems. In this section, we briefly explore its applicability to classical computer vision tasks. Few-shot adaptation functions both as an efficient tuning strategy and as a probe of model competence: if the model succeeds with **very few paired examples**, it indicates that the underlying ability was already internalized during pretraining.

We fine-tune CogVideoX1.5-5B, across tasks using between one and thirty paired examples, maintaining the same architecture, optimization schedule, and hyperparameters as in the main experiments. No auxiliary losses or task-specific modifications are introduced, isolating the contribution of pretrained knowledge.

Table 16: Comparison of CogVideoX1.5 and Qwen3-4B-Instruct-2507 accuracy on *Maze* and *Shortest Path* tasks. Missing values are shown as −.

| n | CogVideoX1.5 | | | Qwen3-4B-Instruct-2507 | | |
|---|---|---|---|---|---|---|
| | Base Maze | Maze Generalization | Shortest Path | Base Maze | Maze Generalization | Shortest Path |
| 3 | 0.015 | − | 0.010 | − | − | − |
| 5 | 0.010 | − | 0.025 | − | − | − |
| 10 | 0.070 | 0.050 | 0.040 | − | − | − |
| 30 | 0.550 | 0.175 | 0.330 | 0.000 | − | 0.010 |
| 50 | 0.760 | 0.355 | 0.420 | 0.005 | 0.000 | 0.010 |
| 100 | 0.940 | 0.590 | 0.700 | 0.005 | 0.000 | 0.050 |
| 300 | 1.000 | 0.755 | 0.860 | 0.115 | 0.020 | 0.155 |
| 500 | 1.000 | 0.885 | 0.910 | 0.195 | 0.060 | 0.320 |
| 1000 | − | 0.865 | 0.945 | 0.500 | 0.335 | 0.500 |
| 3000 | − | 0.815 | 0.960 | 0.710 | 0.375 | 0.640 |
| 5000 | − | 0.940 | 0.975 | 0.925 | 0.525 | 0.770 |

Table 17: Comparison of CogVideoX1.5-5B and Qwen3-4B-Instruct-2507 accuracy on cellular automata rules grouped by Wolfram classes. Missing values are shown as −.

| n | CogVideoX1.5-5B | | | | Qwen3-4B-Instruct-2507 | | | |
|---|---|---|---|---|---|---|---|---|
| | **Class 1** | | | | | | | |
| | R8 | R32 | R128 | R160 | R8 | R32 | R128 | R160 |
| 3 | 0.75 | 0.49 | 0.29 | 0.13 | 0.06 | 0.02 | 0.04 | 0.04 |
| 5 | 0.71 | 0.51 | 0.28 | 0.20 | 0.10 | 0.06 | 0.06 | 0.04 |
| 10 | 0.74 | 0.67 | 0.32 | 0.48 | 0.19 | 0.21 | 0.08 | 0.12 |
| 30 | 0.77 | 0.82 | 0.85 | 0.87 | 0.72 | 0.67 | 0.65 | 0.81 |
| 50 | 0.72 | 0.98 | 0.99 | 0.93 | 0.81 | 0.96 | 0.77 | 0.84 |
| 100 | 1.00 | − | − | − | 0.97 | 0.93 | 0.90 | 0.99 |
| 300 | − | − | − | − | 0.98 | − | − | − |
| | **Class 2** | | | | | | | |
| | R4 | R108 | R170 | R250 | R4 | R108 | R170 | R250 |
| 3 | 0.71 | 0.155 | 0.07 | 0.17 | − | − | − | − |
| 5 | 0.76 | 0.310 | 0.27 | 0.19 | − | − | − | − |
| 10 | 0.74 | 0.415 | 0.87 | 0.27 | − | − | 0.85 | − |
| 30 | 0.85 | 0.640 | 1.00 | 0.59 | 0.72 | 0.47 | 0.99 | 0.52 |
| 50 | 0.93 | 0.785 | 1.00 | 0.90 | 0.82 | 0.82 | 0.98 | 0.86 |
| 100 | − | − | − | − | 0.90 | 0.90 | 1.00 | 1.00 |
| 300 | − | − | − | − | 1.00 | 1.00 | 1.00 | 0.99 |
| | **Class 3** | | | | | | | |
| | R30 | R45 | R90 | R150 | R30 | R45 | R90 | R150 |
| 3 | 0.00 | 0.00 | 0.00 | 0.00 | − | − | − | − |
| 5 | 0.00 | 0.00 | 0.00 | 0.00 | − | − | − | − |
| 10 | 0.00 | 0.00 | 0.00 | 0.00 | − | − | − | − |
| 30 | 0.07 | 0.07 | 0.10 | 0.00 | 0.18 | 0.03 | 0.03 | 0.01 |
| 50 | 0.55 | 0.53 | 0.25 | 0.01 | 0.83 | 0.71 | 0.08 | 0.97 |
| 100 | 0.97 | 1.00 | 0.99 | 0.65 | 0.97 | 0.98 | 0.27 | 0.99 |
| 300 | − | − | − | 0.86 | 1.00 | 1.00 | 0.90 | 1.00 |
| 500 | − | − | − | 0.98 | − | − | − | − |
| | **Class 4** | | | | | | | |
| | R110 | R54 | R62 | R106 | R110 | R54 | R62 | R106 |
| 3 | 0.00 | 0.00 | 0.02 | 0.00 | − | − | − | − |
| 5 | 0.00 | 0.00 | 0.02 | 0.00 | − | − | − | − |
| 10 | 0.00 | 0.01 | 0.03 | 0.00 | − | − | − | − |
| 30 | 0.42 | 0.54 | 0.31 | 0.09 | 0.87 | 0.31 | 0.13 | 0.18 |
| 50 | 0.90 | 0.99 | 0.53 | 0.57 | 0.95 | 0.78 | 0.79 | 0.63 |
| 100 | 1.00 | 1.00 | 0.97 | 0.97 | 1.00 | 0.94 | 0.93 | 1.00 |
| 300 | 1.00 | 1.00 | − | 1.00 | 1.00 | 1.00 | 1.00 | 1.00 |

We explore this setup on several established datasets spanning diverse visual domains, including **NYUv2** Nathan Silberman & Fergus (2012), **ADE20K** Zhou et al. (2017; 2019), **ML-Hypersim** Roberts et al. (2021), **COCO 2017** Lin et al. (2014), and **DreamBooth** Ruiz et al. (2022). These benchmarks cover a wide range of classical computer vision problems, from structured scene understanding to generative image transformation.

Figure 30 illustrates that the model can capture geometric transformations under extreme few-shot conditions. We further show one-shot style transfer in Figure 31.

We also qualitative show this framework can be used to solve some classical computer vision tasks. In Figure 33 we show examples after training with only $n = 30$ samples for *Binary Segmentation* for dogs and *Pose* estimation for humans.

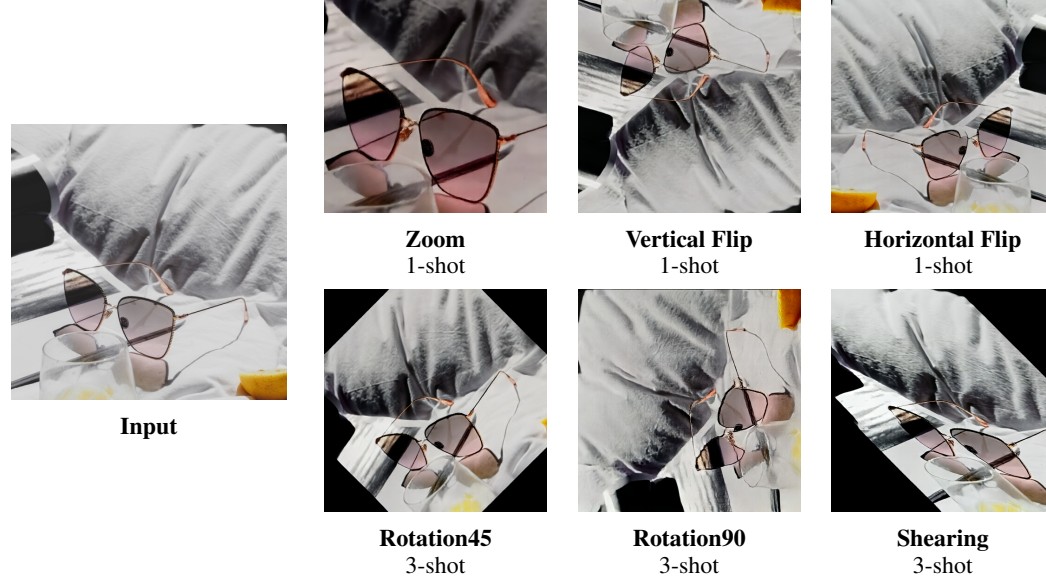

Figure 30: Geometric transformations learned in few-shot setting. Input is shown on the left, with 1-shot results on the top row and 3-shot results on the bottom row.

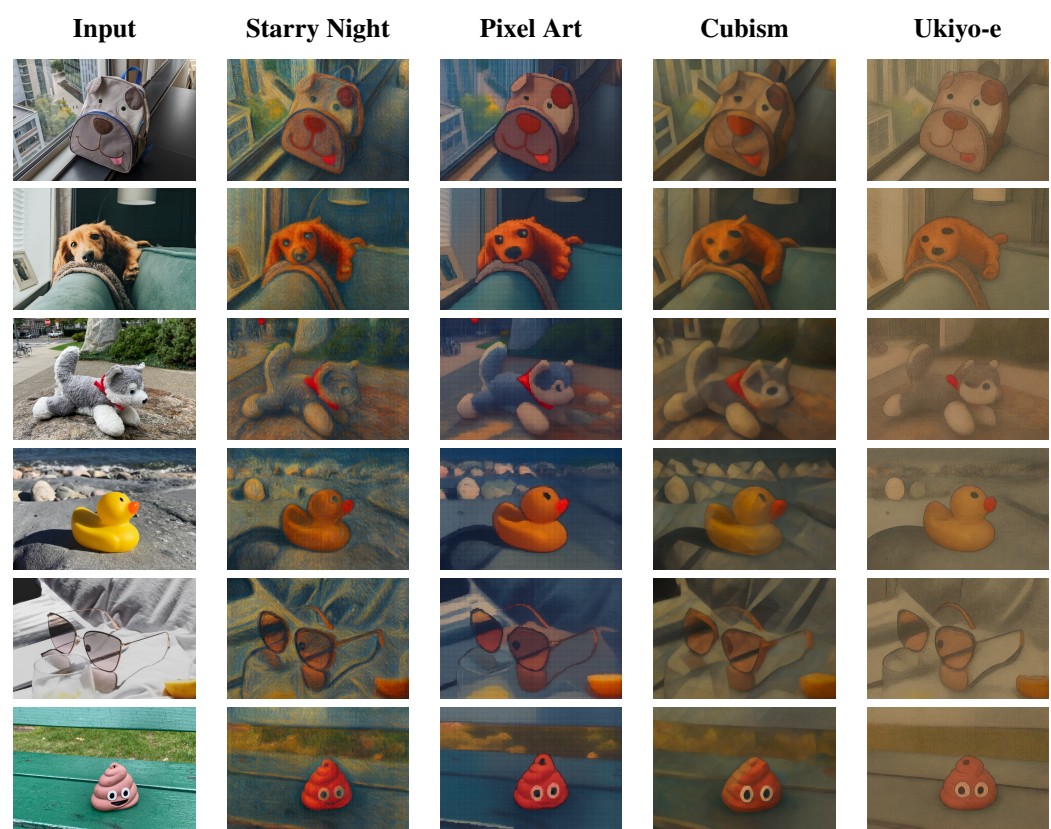

Figure 31: 1-shot style transfer results. The model adapts the input images to distinct artistic styles (*Starry Night*, *Pixel Art*, *Cubism*, and *Ukiyo-e*) using only a single reference example.

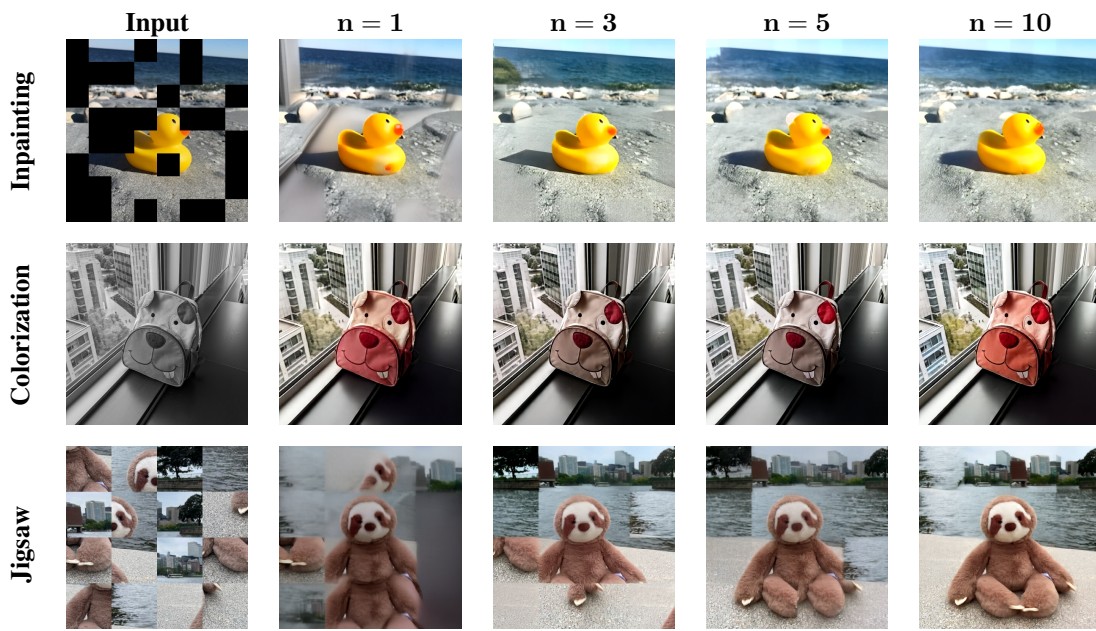

Figure 32: Qualitative results for different tasks (*Inpainting*, *Colorization*, *Jigsaw*) with different numbers of training examples.

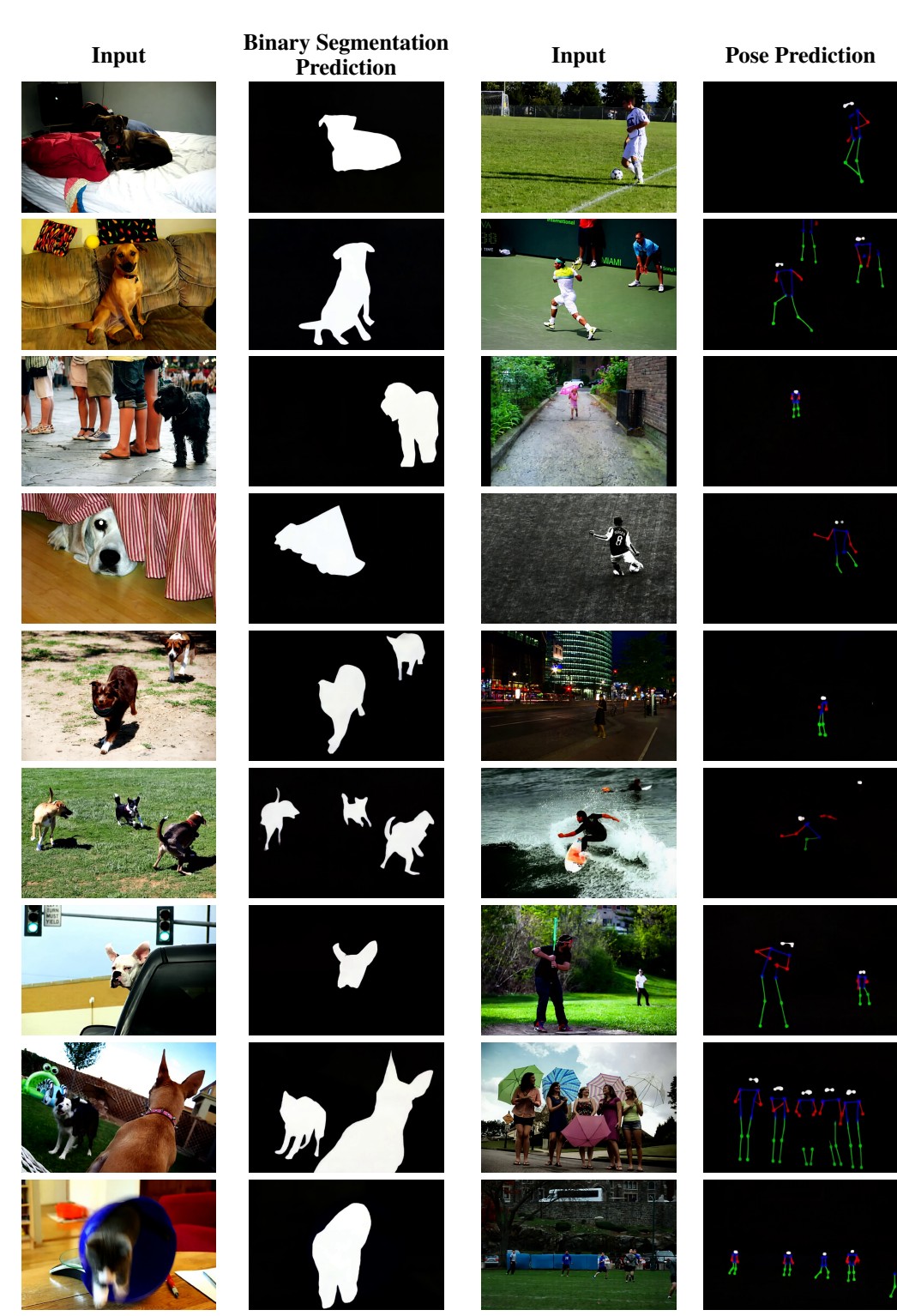

Figure 33: Predictions after finetuning with $n = 30$ samples for *Binary Segmentation* and *Pose*.

**Input**         **Depth Prediction**

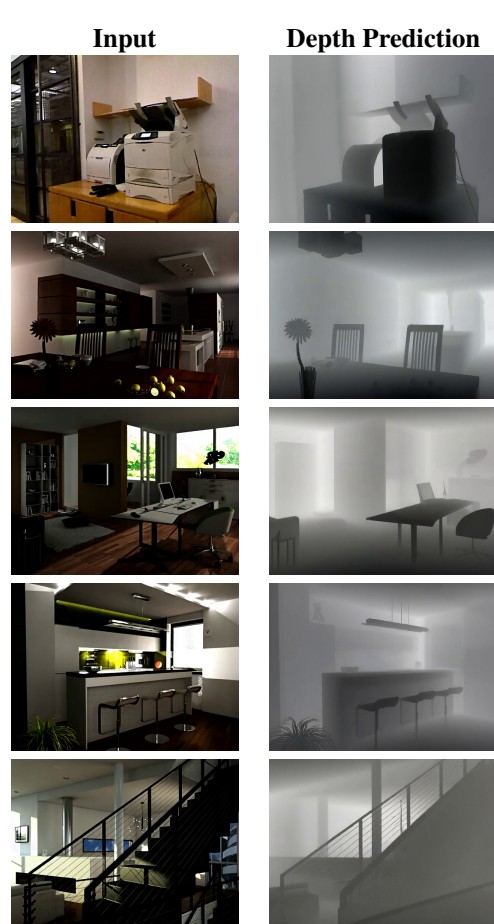

Figure 34: Predictions after finetuning with $n = 30$ samples for *Depth*.

**Input Image**      **Segmentation Prediction**

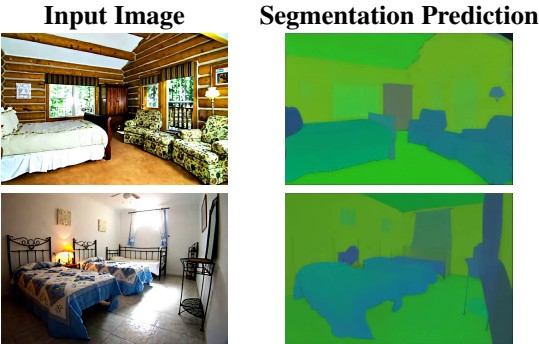

Figure 35: Examples from the *Image → Segmentation* in 1-shot setting for *Chamber*.

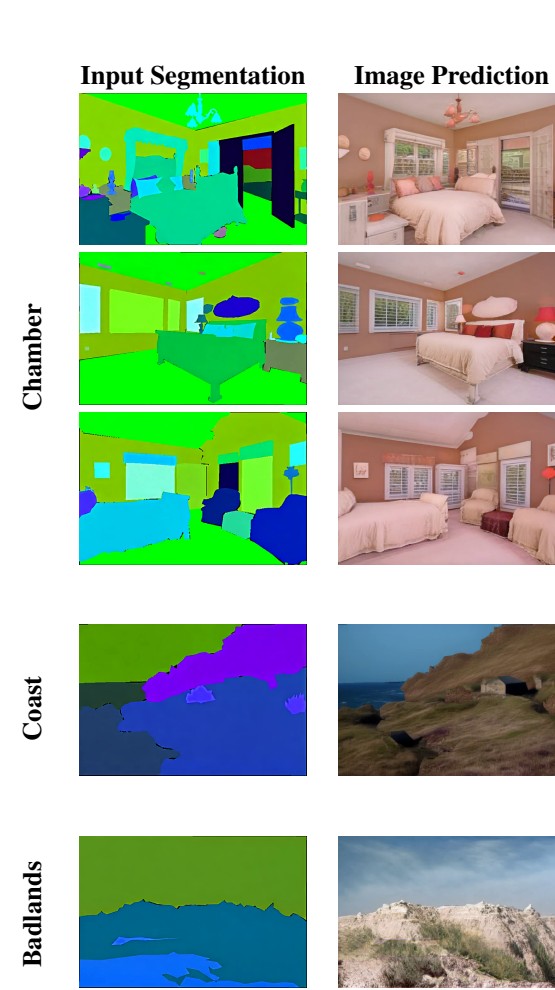

Figure 36: Examples from the *Segmentation → Image* task in the 1-shot setting. Each environment corresponds to a separate 1-shot training: for *Chamber* we train on one chamber and test on others, while for *Coast* and *Badlands* the same protocol applies within their category.

