# OpenReview forum: "Rethinking Visual Intelligence: Insights From Video Pretraining"
_ICLR.cc/2026/Conference — Submitted to ICLR 2026_

### Official Review · Reviewer_W7Aw · 2025-10-17

**Soundness:** 3
**Presentation:** 1
**Contribution:** 2
**Rating:** 2
**Confidence:** 4

**Summary:**

This paper explores Video Diffusion Models (VDMs) as a new direction for pre-training visual intelligence models, aiming to bridge the generalization and data efficiency gap between language models (LLMs) and visual models.

The authors propose a unified framework to rephrase visual tasks as image-to-image as video transition, and systematically compare LLMs and VDM with the same LoRA fine-tuning method across a range of tasks. Results demonstrate that VDMs significantly outperform LLMs in tasks requiring spatial structure understanding, logical or abstract reasoning, and route planning, while also exhibiting higher data efficiency. The paper argues that the inductive bias introduced by spatiotemporal pre-training can contribute to building more general visual foundation models.

**Strengths:**

This article proposes an interesting research direction, rightly noting that visual pre-training (particularly with video) remains underexplored compared to linguistic pre-training. It offers valuable insights, as its title (insights from video pretraining) suggests, and presents experimental evidence that video-pretrained VDMs can generalize more effectively to downstream tasks (e.g., spatial structure understanding, logical or abstract reasoning, and route planning) than language-pretrained models. That's why I give good **Soundness**.

**Weaknesses:**

While I find the core topic of this paper highly interesting, the current manuscript still has significant issues in terms of **sufficiency** and **completeness**.

- **Overly Broad Title & Narrow Focus**: The title is general, while the actual research scope is limited primarily to Video Diffusion Models (VDMs). It should be noted that video foundation models are not exclusively built on diffusion-based approaches—autoregressive (AR) models, for example, represent another important paradigm. The experiments presented are not substantial enough to support the broad claim made in the title. A more accurate title should explicitly reflect that the study focuses on video-diffusion model pre-training?

- **Insufficient Solid Content in Section 3**: The content of Section 3 reads as overly conceptual, covering mostly established background knowledge. Dedicating two full pages to such content does not effectively advance the paper's argument (There are many cases where a single sentence is a standalone paragraph, seeming unnecessary to spend so much space?). It would be more valuable to incorporate additional experimental results and analyses from the appendix into the main body. In its current form, this section gives the impression of a lack of refinement, which is the main reason for my low score in **Presentation**.

- **Limited Scope of Comparative Evaluation**: The study only demonstrates VDM's advantages over LLMs in tasks that can be naturally framed as image-to-image transitions. However, it remains unclear how VDMs would be applied to tasks that are not easily represented as image inputs—such as mathematical reasoning, code generation, or knowledge tasks. What would the input form be for such cases? Would VDMs still maintain an advantage? Most video foundation models start with image foundation models. Since the title mentions **rethinking visual intelligence**, why is it limited to discussing video pre-trained models and not including pure image pre-trained models? The experimental part conducted in the article is not sufficient to support the title of this article. In addition, why do not compare with VLM (image-text input, text out), and t2i/t2v model with text input, image ouput?

**In summary, I believe the current manuscript requires considerable improvement and comes across as somewhat rushed. With substantial revision and expansion, I think it may be a good article**.

**Questions:**

See Weakness.

Another point concern is that how to determine the correctness of the VDM's output image, specifically how it is evaluated against the ground truth. I did not find the corresponding descriptions for this in the main text.

---

> ### Author Response · Authors · 2025-11-14
>
> We thank the reviewer for the detailed feedback. We address the main concerns below.
>
> **Overly Broad Title & Narrow Focus**
>
> Our goal is to study how spatiotemporal pretraining shapes inductive biases for visual intelligence. Currently, video diffusion models are a class of large video models that consistently demonstrate high-fidelity generation, strong temporal consistency, and stable conditioning under partial visual inputs. Autoregressive video models are an interesting field of study, but they have not yet reached the level of their diffusion counterparts. We expect the same behaviour to emerge in autoregressive video models as they mature. Since we believe the mechanisms we analyse are tied to modality-aligned pretraining rather than to a specific architecture, we think the conclusions remain relevant.
>
> **On Section 3 and the concern about presentation**
>
> Section 3 sets out a symmetric comparison framework for video diffusion models and language models. Although parts of it revisit established ideas, this foundation is necessary for explaining how we maintain fairness across modalities. The section details the alignment between image-based and text-based representations, the LoRA protocol, and the evaluation setup. These elements are crucial for interpreting the results and for showing why the comparison is sound.
>
> We regret that the presentation did not connect with the reviewer. Our goal in this section was to provide a clear explanation that enables readers to understand the parallelism between fine-tuning in language models and video diffusion models, as well as the alignment of their respective modalities, so that the methodology is not misunderstood. We appreciate the feedback and will strive to condense this into a more concise presentation.
>
> **On the scope of comparison and the request to include tasks that are not naturally visual**
>
> The reviewer asks why tasks such as mathematical reasoning, coding, or knowledge retrieval tasks are not included. Some of these tasks do not primarily involve visual intelligence. The goal of this work is to demonstrate that video models can be a promising direction towards universal problem solvers in the visual domain, **rather than competing with LLMs at every task.**
>
> Regarding image diffusion models, we refer the reviewer to our response to Reviewer Zamz. There is currently no established few-shot adaptation protocol for image diffusion models that preserves a frozen backbone setting. Existing approaches require architectural changes or instance-level specialization that break the symmetry of comparison. Attempting to adapt them would effectively turn them into video models. For this reason, we focus on video diffusion models that incorporate conditional image-to-image prediction as a native capability.
>
> Vision language models (VLM) are explored in the appendix ("PITFALLS OF VISION LANGUAGE MODELS"). Their encoders often collapse fine spatial information in favour of semantics, which limits their performance on structural visual tasks. For T2I or T2V, there is, as far as we know, no principled way to integrate them into our evaluation framework **in a fair way**. We are open to suggestions on how to adapt them while preserving fairness.
>
> **On evaluation correctness for video diffusion model outputs**
>
> All visual outputs are treated as RGB grids and converted back to their symbolic representation using the inverse of the same deterministic rendering function used to produce the image.
>
> We appreciate the reviewer’s careful evaluation. We hope that once the concerns are fully addressed, the reviewer will feel confident in raising the score.

---

> ### Comment · Reviewer_W7Aw · 2025-11-26
> **Reply to Author**
>
> Thanks for the authors’ response.
> ﻿
> As you mentioned, experiments were only conducted on VDM, so the title may specifically emphasize VDM more. Video pretraining is a broad topic, and conducting experiments on more video pretraining paradigms, such as video autoregressive (AR) models and hybrid models combining AR-and-Diff, would be more comprehensive and convincing (although these paradigms may not be as strong as VDM).
> ﻿
> I recommend that the authors refine Section 3, as it seems to introduce common knowledge in this field (most content in Sec. 3). The section is divided into many paragraphs, and it appears that devoting so much space to introducing this content is not particularly valuable.

---

### Official Review · Reviewer_Zamz · 2025-11-01

**Soundness:** 2
**Presentation:** 2
**Contribution:** 2
**Rating:** 4
**Confidence:** 3

**Summary:**

The authors evaluate video diffusion models and LLMs on visual tasks like ARC-AGI, path planning, sudoku etc. using a LoRA based finetuning setup, to demonstrate that video diffusion models are more data efficient than LLMs when it comes to visual reasoning tasks.

**Strengths:**

1. The authors have curated a set of interesting visual tasks to benchmark the spatial reasoning capacity of VDMs and LLMs from cellular automata to visual games.
2. The ARC-AGI results are quite novel and timely and highlight drawbacks of current LLMs.

**Weaknesses:**

1. It seems all the visual tasks require spatial reasoning, not spatio-temporal reasoning, which begs the question why not evaluate image diffusion models as well instead of video diffusion models where the uathors practically discard the temporally intermediate frames generated by the model, essentially not using/evaluating the temporal reasoning capacity of these models.

2. The data efficiency plots compare cog-x with qwen without controlling for pre-training FLOPs/data-volume. This is a very important factor that can determine baseline model performance and should be reported.

3. The authors need to show results for more than one VDM and LLM across all these tasks in order to make general claims about model families.

4. The authors also need to demonstrate scaling behaviors of these VDMs, showing improvement in data efficiency/visual reasoning performance, with increase in model params/pretraining FLOPs etc. to support the claim that VDMs can become foundational vision models.

5. Finally, for the sake of completeness, the authors should also analyse VDMs vs LLMs for sequence reasoning tasks, to give a full picture of the strengths and weaknesses of these large vision/language models.


Overall the paper tries to do interesting and relevant analysis of VDMs and LLMs but still lacks crucial results and experiments.

**Questions:**

see weaknesses

---

> ### Author Response · Authors · 2025-11-14
>
> We thank the reviewer for the detailed feedback. We address the raised concerns below.
>
> 1. Exploring image diffusion models in a few example settings would certainly be valuable. Unfortunately, we are unaware of any native method for prompting image models with initial images in a few fine-tuning settings without effectively "converting them into video models" (by means of some additional pre-training I2I). This makes the comparison difficult to conduct.
>
> 2. Regarding the pre-training scale, the Qwen3 technical report states that the model is trained on approximately thirty-six trillion tokens, followed by a final stage on high-quality text. For CogVideoX, the published data description indicates the use of approximately thirty-five million single-shot clips, each averaging 6 seconds in length, together with two billion images filtered from LAION and 700M COYO. The precise token count cannot be derived directly because it depends on the resolutions used throughout training, although it is reasonable to assume that most training occurred at lower resolutions before a final high-resolution phase. Based on this, the total number of effective tokens is expected **to be safely below the thirty-six trillion tokens used for the LLM**. We were unable to locate detailed information for CogVideoX1.5, although it is likely of a similar order of magnitude. **We believe the domain of the pre-training data is a much more influential factor for these tasks than the absolute token count.**
>
> Qwen technical report: https://arxiv.org/pdf/2505.09388
> CogVideoX technical report: https://arxiv.org/pdf/2408.06072
>
> 3. We have comparisons on ConceptARC, which include additional LLM and VDM families, and observed consistent trends. Extending this broader comparison to every task would require substantial computational resources. If the reviewer considers this essential, we can attempt to include one additional model family (one LLM and another VDM) for the camera-ready version for all the tasks. Performing this during the rebuttal period is not computationally feasible, although we expect the outcomes to remain aligned with the existing observations.
>
> 4. Scaling effects are visible in ConceptARC, where Wan2.1 shows markedly stronger performance, supporting the expectation that larger VDMs improve on visual tasks.
>
> 5. This direction is interesting but falls outside the scope of our work. Our method tunes VDMs for Image-to-Image tasks, rather than Image-to-Video tasks, which are necessary for sequence modeling; although we expect VDMs to perform reasonably well in this context. The primary bottleneck lies in the LLMs. Handling sequence tasks would require the LLM to generate multiple JSON objects, which quickly inflates the token count and results in much higher computational cost. For the VDM, this also adds significant overhead, although it is likely manageable given the computational constraints. Moreover, the LLM already encounters difficulties with single Image-to-Image understanding tasks, so achieving stable performance on sequence tasks would probably demand a substantially larger number of samples.
>
> We value the reviewer’s thoughtful assessment. We hope that after the points raised are resolved, the reviewer will reconsider the score.

---

### Official Review · Reviewer_LGZ3 · 2025-11-01

**Soundness:** 3
**Presentation:** 3
**Contribution:** 3
**Rating:** 4
**Confidence:** 4

**Summary:**

The paper compares video diffusion models (VDMs) and large language models (LLMs) under a symmetric, frozen-backbone + LoRA protocol: VDMs perform image-to-image prediction by reframing each input–output grid as a short transition video with discrete interpolation and a neutral fixed text embedding, while LLMs do JSON-to-JSON sequence prediction. Evaluation spans ARC-AGI/ConceptARC, visual games, route planning, and cellular automata with sample-efficiency curves; results show that VDMs are often more sample-efficient on spatial/temporal structure, supporting the claim that video pretraining offers a powerful foundation for visual intelligence.

**Strengths:**

- Novel Hypothesis and Reframing: The paper's core strength is its originality in reframing VDMs as general problem-solvers rather than just generators. The hypothesis that spatiotemporal inductive biases are key to visual intelligence is a significant and insightful contribution.
- Focus on Data Efficiency: The evaluation wisely focuses on skill acquisition efficiency instead of just final SOTA performance. This provides much deeper evidence for the VDM's superior learning properties in low-data regimes.

**Weaknesses:**

1. Fundamental Asymmetry in Task Representation and Modality: The comparison's fairness is highly questionable due to a core mismatch in task modalities.
(1)	The LLM must perform a text-to-text translation on JSON-serialized grids , while the VDM performs a direct pixel-to-pixel mapping. These two representations have fundamentally different information densities, processing complexities, and inherent difficulties. (For example, given a 5x5 grid structure, VDM needs to process an image of 256x256 pixels, while LLM needs to process 25 numbers.)
(2)	The LLM faces a "dual burden" of mastering a complex JSON syntax in addition to the task's core logic. Could the LLM's poor data efficiency be a result of this syntactic and representational overhead, rather than a true failure of its inductive bias for logic?
2. Limited Task Scope and Ecological Validity: The paper makes strong claims about visual intelligence and visual foundation models , but the evaluation is confined to a curated set of synthetic, grid-based tasks with explicit human-defined rules (e.g., ARC, Sudoku, Mazes). This success on toy problems, which are highly amenable to grid-based serialization, does not guarantee the VDM's advantage will generalize to the ambiguity, noise, and implicit physical rules of real-world perception.
3. Low Absolute Performance: Despite relative efficiency gains, the VDM's low absolute accuracy on key abstract tasks (like ARC-AGI) suggests it also has fundamental limitations in abstract generalization. This low ceiling challenges the claim that this bias is a sufficient solution.
4. Un-ablated Text Embedding: The VDM is conditioned on a "neutral fixed text embedding" ($e_{text}$). Without an ablation study, it is unclear if this provides a crucial task hint, acting as an unfair advantage for the VDM over the LLM, which received no such meta-prompt.

Open to increasing my score, provided my concerns are addressed.

**Questions:**

Please refer to the questions mentioned in 'weaknesses' section.

---

> ### Author Response · Authors · 2025-11-14
>
> We thank the reviewer for the careful reading and constructive concerns. We address each point below.
>
> 1. Regarding the modality asymmetry, the difference in task representations is actually central to our argument. There are tasks for which visual representations are inherently required, for which these pretrained biases will be useful. Current VLM approaches attempt to bridge this gap, but as we show in the small ablation in the appendix, existing VLM architectures struggle with fine-grained spatial information and perform no better than LLMs (this line of work has also been explored by other authors). We do not view this as a limitation of our method but rather as evidence that the choice of modality is part of the hypothesis itself.
>
> We also experimented with different textual encodings for the LLM, including JSON and grid text, and found broadly similar behavior. We therefore used JSON because it is the standard representation used in ARC-AGI evaluations.
>
> 2.1. We do not think the question of extra syntactic burden presents a meaningful obstacle for the model. Modern large language models receive extensive pretraining on JSON and many other structured formats, so the syntax does not create any major difficulty. It is no more challenging than a VDM that is trained mainly on natural images and later asked to interpret two-dimensional colored grids. As noted above, we tried alternative text formats and observed no meaningful differences. Moreover, the vast majority of few-shot LLM approaches to ARC-AGI adopt JSON as the canonical format, and this has not been reported as a major impediment in the literature.
>
> 2.2. On task scope and ecological validity, we agree that the benchmarks we use are targeted and simplified. This choice is intentional because the comparison to LLMs requires tasks that can be reasonably serialized into a text format. We view this as establishing a controlled testbed for the symmetric frozen backbone plus LoRA protocol, rather than claiming full ecological validity (which we have indicated as a limitation). Extending the VDM-based approach to more realistic visual tasks is an important direction for future work, though such tasks would not permit a symmetric comparison to text-only LLMs. We have provided a brief initial exploration in the last section of the Appendix, which already shows some promise.
>
> 2.3. On absolute performance, especially on ARC-AGI, we actually view the results as strong given the constraints. Current top ARC-AGI systems typically leverage much larger LLMs with a chain of thought (CoT) or more aggressive test time adaptation, including example swapping (for IC examples) and augmentation, after which they do some kind of ensembling. Smaller models tend to be heavily pre-trained on large ARC-like corpora, such as ARC Heavy 200k, and even modifying the tokenizer to ingest grid tokens directly. None of these are compatible with our setting because they break the backbone protocol and obscure the comparison, and are not natively extensible to different tasks. We purposefully avoid these forms of external assistance, including data augmentation, to isolate the efficiency of skill acquisition. Under these constraints, we consider the VDM results competitive and informative.
>
> 2.4. On the text embedding used for the VDM, we apologize for not being explicit. The conditioning text was the fixed string “A static image being processed. No movement.” and the embedding was frozen throughout. We also tested an empty text embedding and found no real difference. This embedding does not encode any task information and was included only because the base VDM architecture expects a text input.
>
> We hope these clarifications resolve the concerns, and we appreciate the reviewer’s openness to increasing the score. We hope that once the issues highlighted are clarified, the reviewer will update the score.

---

### Official Review · Reviewer_ykQ1 · 2025-11-02

**Soundness:** 2
**Presentation:** 2
**Contribution:** 2
**Rating:** 2
**Confidence:** 3

**Summary:**

The authors investigate whether VDMs can serve as a foundation for visual intelligence, like how LLM did for NLP. They use spatiotemporal pretraining to enhance generalization and data efficiency and provide experimental results. They also curated 3 synthetic benchmarks to test those. For controlled experiments, both were adapted with LoRA fine-tuning while keeping backbones frozen. Each model receives tasks in its natural modality.

**Strengths:**

1. The motivation of this work is good; the questions the authors raised deserve to have a work to study them.
2. The curated tasks of interest are interesting; they designed synthetic tasks to test them
3. They show some results that pretraining on VDM modality specific tasks would improve its downstream performances.

**Weaknesses:**

1. The synthetic tasks are too simplified to be indicative to downstream or other tasks performance. If the authors can show some downstream application enhancements, even just 1 example, then it would be more convincing.
2. The authors compare LLM and VDM, which are two different architectures. There may be transformer-based video LLMs available, such as VideoPoet and VAR, among others. I am sure there are also open-sourced alternatives that are more suitable for these comparisons.
3. The ablations of experiments can be more extensive to be convincing, e.g., add 1 or 2 more families of LLMs and VDMs, add different sizes of those LLMs/VDMs, or, since the authors use LoRA, maybe there can be one more config, etc.

**Questions:**

See weaknesses. Also, maybe the authors can compare different family or Visual foundation models, e.g., VDMs vs transformer-based VLMs. I wonder how those results would fare?

---

> ### Author Response · Authors · 2025-11-14
>
> We thank the reviewer for the thoughtful assessment and for highlighting both the motivation of the study and the interest of the curated tasks. We address each point below.
>
> **On the simplicity of the synthetic tasks and lack of downstream demonstrations**
> Our work focuses on controlled settings to examine how modality-aligned pretraining and inductive biases influence skill acquisition. The grid-based tasks provide a neutral structural interface that allows fair comparison between LLMs and VDMs. As mentioned in the paper, extending these insights to naturalistic or embodied domains is future work, but those settings do not offer a principled or comparable way to evaluate LLMs since no serialization preserves the required structural detail.
> We explored more natural image-to-image examples in Appendix G, where VDMs behave well, but an LLM baseline is not feasible under a comparable setup.
>
> We also note that ARC-AGI is widely regarded as the gold standard benchmark for abstract generalization, despite its simplicity in representation.
>
> **On architectural differences and the availability of transformer-based video LLMs**
> While architectures differ, our observations suggest that the decisive factor in performance is the pretraining modality rather than architectural specifics. To the best of our knowledge, diffusion-based LLMs have not shown advantages over autoregressive LLMs on tasks like ARC-AGI. For this reason, we selected a well-established LLM family that is known to be competitive.
>
> **On ablations and additional model families**
> We have comparisons for ConceptARC, including more LLM and VDM families, and observed consistent trends.
>
> Broader sweeps across the full set of tasks could be valuable, but we were unable to complete them during the rebuttal period due to computational constraints. The LoRA configurations used were selected based on common practice and validated in preliminary trials; we did not observe substantial sensitivity to their exact values. If the reviewer believes that adding one more LLM and VDM family would be essential for the final version, we are prepared to include this in the camera-ready version, although we cannot perform it within the rebuttal window due to computational constraints.
>
> **On comparisons with visual foundation models such as VLMs**
> An analysis of VLMs appears in the appendix "PITFALLS OF VISION LANGUAGE MODELS". Both our findings and prior work indicate that their visual encoders often fail to retain the structural spatial detail required for the tasks in this study, focusing on semantics. In our experiments, they do not surpass LLMs. Moreover, most VLMs training is an additional tuning phase with a visual encoder **on top of a pre-existing, pre-trained LLM**. For this reason, they were not included in the main paper.
>
> We hope these clarifications address the reviewer’s concerns and help contextualize the scope of the work. We hope that after the noted points are properly addressed, the reviewer will be willing to revise the score.

---

### Meta-Review · Area_Chair_6dJc · 2026-01-05

**Summary:**

This paper explores the hypothesis that Video Diffusion Models (VDMs), due to their spatiotemporal pretraining, possess superior inductive biases for visual intelligence compared to Large Language Models (LLMs). The authors evaluate this by reframing grid-based tasks (ARC-AGI, Sudoku, route planning) as image-to-image video transitions for VDMs and JSON-to-JSON translations for LLMs. While the paper finds that VDMs are more data-efficient in these specific settings, all four reviewers raised significant concerns regarding the fundamental fairness of the comparison, the limited ecological validity of the toy tasks, and the mismatch between the paper’s broad title and its narrow focus.

**Reviewer Concerns:**

The authors’ rebuttal attempted to address several points, but fundamental issues remain outstanding:

- Fundamental Asymmetry (LGZ3, ykQ1): The core concern is the "apples-to-oranges" nature of comparing pixel-level generation to JSON sequence prediction. Reviewers argue that the LLM's poor data efficiency may stem from the syntactic overhead of JSON and the loss of spatial structural information during serialization, rather than a lack of "visual intelligence." The authors argue this asymmetry is the point of the study, but I agree with reviewers that this setup fails to isolate "pretraining bias" from "modality encoding" effects.

- Task Validity and Scope (W7Aw, Zamz, LGZ3): The evaluation is confined to synthetic, grid-based "toy" problems. Reviewer W7Aw noted that the paper makes sweeping claims about "Visual Intelligence" and "Foundation Models" that are not supported by evidence from natural images or complex, real-world visual reasoning. The qualitative results in the Appendix do not sufficiently bridge this gap.

- Mismatch of Mechanism (Zamz): Reviewer Zamz pointed out that the selected tasks are essentially static and do not require temporal reasoning. Using a Video Diffusion Model—and effectively discarding the intermediate temporal frames—to solve static grid transformations suggests that the "video" aspect of the pretraining is not being rigorously tested against the tasks' requirements.

- Low Absolute Performance (LGZ3): Despite the claimed efficiency gains, the absolute accuracy on benchmarks like ARC-AGI remains very low (16.75%). This suggests that the proposed approach, while an interesting probe, does not yet provide a viable path toward competitive visual reasoning.

**Reviewer Scores:**

If a full discussion period had occurred, I believe the scores would have remained firmly in the "Reject" category:

- Reviewer ykQ1 (2): Likely to remain at a 2. The author's response did not resolve the concern that comparing two fundamentally different architectures on a single serialization format is insufficient for a general claim.

- Reviewer Zamz (2): Likely to remain at a 2. The author’s admission that the VDM uses a "static image" prompt and that temporal reasoning is not strictly required by the tasks reinforces this reviewer's primary criticism.

- Reviewer W7Aw (2): Likely to remain at a 2. While the reviewer acknowledged the "Soundness" of the research direction, the "Presentation" and "Sufficiency" issues (toy tasks, conceptual density of Section 3, broad title) were not adequately addressed by the minor revisions offered.

- Reviewer LGZ3 (4): Might have trended toward a 3. While the reviewer was open to the hypothesis, the lack of a "bridge" between the toy results and real-world visual intelligence makes the contribution too narrow for a top-tier conference.

---

### Decision · Program_Chairs · 2026-01-26

Reject